# Hierarchical Optimal Transport for Multimodal Distribution Alignment

**John Lee**[†*]**, Max Dabagia**[†]**, Eva L. Dyer**[†‡§]**, Christopher J. Rozell**[†§]
[†]School of Electrical and Computer Engineering,
[‡]Coulter Department of Biomedical Engineering
Georgia Institute of Technology, Atlanta, GA, 30332 USA
{john.lee, maxdabagia, evadyer, crozell}@gatech.edu

## Abstract

In many machine learning applications, it is necessary to meaningfully aggregate, through alignment, different but related datasets. Optimal transport (OT)-based approaches pose alignment as a divergence minimization problem: the aim is to transform a source dataset to match a target dataset using the Wasserstein distance as a divergence measure under alignment constraints. We introduce a hierarchical formulation of OT which leverages clustered structure in data to improve alignment in noisy, ambiguous, or multimodal settings. To solve this numerically, we propose a distributed ADMM algorithm that exploits the Sinkhorn distance, thus it has an efficient computational complexity that scales quadratically with the size of the largest cluster. When the transformation between two datasets is unitary, we provide performance guarantees that describe *when* and *how well* cluster correspondences can be recovered with our formulation, and then describe the worst-case dataset geometry for such a strategy. We apply this method to synthetic datasets that model data as mixtures of low-rank Gaussians and study the impact that different geometric properties of the data have on alignment. Next, we applied our approach to a neural decoding application where the goal is to predict movement directions and instantaneous velocities from populations of neurons in the macaque primary motor cortex. Our results demonstrate that when clustered structure exists in datasets, and is consistent across trials or time points, a hierarchical alignment strategy that leverages such structure can provide significant improvements in cross-domain alignment.

## 1 Introduction

In many machine learning applications, it is necessary to meaningfully aggregate, through alignment, different but related datasets (e.g., data across time points or under different conditions or contexts). Alignment is an important problem at the heart of transfer learning [1, 2], point set registration [3, 4, 5], and shape analysis [6, 7, 8], but is generally NP hard. In recent years, distribution alignment methods that use optimal transport (OT) to quantify similarity between two distributions have increased in popularity due to their attractive mathematical properties and impressive performance in a variety of tasks [9, 10]. However, using OT to solve unsupervised distribution alignment problems that must simultaneously match two datasets' distributions (using OT) while also learning a transformation between their latent spaces, is extremely challenging, especially when the data has complicated multi-modal structure. Leveraging additional structure in the problem is thus necessary to regularize OT and constrain the solution space.

Here, we leverage the fact that heterogeneous datasets often admit *clustered* or *multi-subspace* structure to improve OT-based distribution alignment. Our solution to this problem is to simultaneously

---

[*]JL is currently with DSO National Laboratories of Singapore.
[§]Equal contributing senior authors.

estimate the cluster alignment across two datasets using their local geometry, while also solving a global alignment problem to meld these local estimates. While it is advantageous to regularize the OT problem with known cluster pairings [10, 11], we are instead concerned with the substantially harder unsupervised setting where such information is missing. We introduce a hierarchical formulation of OT for clustered and multi-subspace datasets called *Hierarchical Wasserstein Alignment (HiWA)*[3].

We empirically show that when data are well approximated with Gaussian mixture models (GMMs) or lie on a union of subspaces, we may leverage existing clustering pipelines (e.g., sparse subspace clustering [12] [13]) to improve alignment. When the transformation between datasets is unitary, we provide analyses that reveal key geometric and sampling insights, as well as perturbation and failure mode analyses. To solve the problem numerically, we propose an efficient distributed ADMM algorithm that also exploits the Sinkhorn distance, thus benefiting from efficient computational complexity that scales quadratically with the size of the largest cluster.

To test and benchmark our approach, we applied it to synthetic data generated from mixtures of low-rank Gaussians and studied the impact of different geometric properties of the data on alignment to confirm the predictions of our theoretical analysis. Next, we applied our approach to a neural decoding application where the goal is to predict movement directions from populations of neurons in the macaque primary motor cortex. Our results demonstrate that when clustered structure exists in neural datasets and is consistent across trials or time points, a hierarchical alignment strategy that leverages such structure can provide significant improvements in unsupervised decoding from ambiguous (symmetric) movement patterns. This suggests OT can be applied to a wider range of neural datasets, and shows that a hierarchical strategy avoids local minima encountered by a global alignment strategy that ignores clustered structure.

## 2   Background and related work

**Transfer learning and distribution alignment.**   A fundamental goal in transfer learning is to aggregate related datasets by learning a mapping between them. We wish to learn a transformation $T \in \mathcal{T}$, where $\mathcal{T}$ refers to some class of transformations that aligns distributions under a notion of probability divergence $\mathcal{D}(\cdot|\cdot)$ between a target distribution $\mu$ and a reference (source) distribution $\nu$:

$$\min_{T \in \mathcal{T}} \mathcal{D}(T(\mu)|\nu). \tag{1}$$

Various probability divergences have been proposed in the literature, such as Euclidean least-squares (when data ordering is known) [14, 15, 16], Kullback-Leibler (KL) [17], maximum mean discrepancy (MMD) [18, 19, 20, 21], and the Wasserstein distance [10], where trade-offs are often statistical (e.g., consistency, sample complexity) versus computational. Alignment problems are ill-posed since the space of $\mathcal{T}$ is large, so *a priori* structure is often necessary to constrain $\mathcal{T}$ based on geometric assumptions. Compact manifolds like the Grassmann or Stiefel [22, 23] are primary choices when little information is present, as they preserve isometry. Non-isometric transformations, though richer, demand much more structure (e.g., manifold or graph structure) [24, 25, 26, 27, 10].

**Low-rank and union of subspaces models.** Principal components analysis (PCA), one of the most popular methods in data science, assumes a *low-rank* model where the top-$k$ principal components of a dataset provide the optimal rank-$k$ approximation under an Euclidean loss. This has been extended to robust (sparse errors) settings [12], and multi- (union of) subspaces settings where data can be partitioned into disjoint subsets where each subset of data is locally low-rank [28]. Transfer learning methods based on subspace alignment [29, 30, 31] work well with zero-mean unimodal datasets, but struggle on more complicated modalities (e.g., Gaussian mixtures or union of subspaces) due to a mixing of covariances. Related to our work, [32] performs multi-subspace alignment by greedily assigning correspondences between subspaces using chordal distances; this however discards valuable information about a distribution's shape.

**Optimal transport.**   Optimal transport (OT) [33] is a natural type of divergence for registration problems because it accounts for the underlying geometry of the space. In Euclidean settings, OT gives rise to a metric known as the Wasserstein distance $\mathcal{W}(\mu, \nu)$ which measures the minimum effort required to "displace" points across measures $\mu$ and $\nu$ (understood here as empirical point clouds). Therefore, OT relieves the need for kernel estimation to create an overlapping support of the measures

$\mu, \nu$. Despite this attractive property, it has both a poor numerical complexity of $O(n^3 \log n)$ (where $n$ is the sample size) and a dimension-dependent sample complexity of $O(n^{-1/d})$, where the data dimension is $d$ [34, 35]. Recently, an entropically regularized version of OT known as the Sinkhorn distance [36] has emerged as a compelling divergence measure; it not only inherits OT's geometric properties but also has superior computational and sample complexities of $O(n^2)$ and $O(n^{-1/2})^4$, respectively. It has also become a versatile building block in domain adaptation [10, 38]. Prior art [10] has largely exploited the OT's push-forward as the alignment map since this map minimizes the OT cost between the source and target distributions while allowing a priori structure to be easily incorporated (e.g., to preserve label/graphical integrity). Such an approach, however, is fundamentally expensive when $d \ll n$ since the primary optimization variable is a large transport coupling (i.e., $\mathbb{R}^{n \times n}$), while in reality the alignment mapping is merely $\mathbb{R}^d \mapsto \mathbb{R}^d$. Moreover, it assumes that the source and target distributions are close in terms of their squared Euclidean distance (i.e., an identity transformation), but this does not generally hold between arbitrary latent spaces.

**Hierarchical OT and related work.** The idea of learning an affine or unitary transformation to align datasets with an OT-based divergence has previously been studied in [39, 40, 41], a problem known as OT Procrustes. However, these methods don't use problem-specific or clustered structure in data. Hierarchical OT is a recent generalization of OT [42, 43, 44] that is an effective and efficient way of injecting structure into OT but it has never been used to *jointly* solve alignment problems – our work represents a first attempt at doing so. Thus, a key contribution of this paper is putting both of these two ingredients together to develop a scalable strategy that leverages multimodal structure in data solve the OT Procrustes problem.

## 3  Hierarchical Wasserstein alignment

**Preliminaries and notation.** Consider clustered datasets $\{\boldsymbol{X}_i \in \mathbb{R}^{D \times n_{x,i}}\}_{i=1}^S$ and $\{\boldsymbol{Y}_j \in \mathbb{R}^{D \times n_{y,j}}\}_{j=1}^S$ whose clusters are denoted with the indices $i, j$ and whose columns are treated as $\mathbb{R}^D$ embedding coordinates. The number of samples in the $i$-th ($j$-th) cluster of dataset $\boldsymbol{X}$ (dataset $\boldsymbol{Y}$) is given by $n_{x,i}$ ($n_{y,j}$). We express the empirical measures of clusters $\boldsymbol{X}_i$ and $\boldsymbol{Y}_j$ as $\mu_i := \frac{1}{n_{x,i}} \sum_{k=1}^{n_{x,i}} \delta_{\boldsymbol{X}_i(k)}$ and $\nu_j := \frac{1}{n_{y,j}} \sum_{l=1}^{n_{y,j}} \delta_{\boldsymbol{Y}_j(l)}$, respectively, where $\delta_{\boldsymbol{x}}$ refers to a point mass located at coordinate $\boldsymbol{x} \in \mathbb{R}^D$. The squared 2-Wasserstein distance between $\mu_i$ and $\nu_j$ is defined as

$$\mathcal{W}_2^2(\mu_i, \nu_j) := \min_{\boldsymbol{Q} \in \mathcal{U}(n_{x,i}, n_{y,j})} \sum_{k=1}^{n_{x,i}} \sum_{l=1}^{n_{y,j}} Q(k,l) \left\| \boldsymbol{X}_i(k) - \boldsymbol{Y}_j(l) \right\|_2^2$$

where $\boldsymbol{Q}$ is a doubly stochastic matrix that encodes point-wise correspondences (i.e., the $(k, l)$-th entry describes the flow of mass between $\delta_{\boldsymbol{X}_i(k)}$ and $\delta_{\boldsymbol{Y}_j(l)}$), $\boldsymbol{X}_i(k)$ is the $k$-th column of matrix $\boldsymbol{X}_i$, and the constraint $\mathcal{U}(m,n) := \{\boldsymbol{Q} \in \mathbb{R}_+^{m \times n} : \boldsymbol{Q} \mathbb{1}_n = \mathbb{1}_m/m, \boldsymbol{Q}^\top \mathbb{1}_m = \mathbb{1}_n/n\}$ refers to the *uniform* transport polytope (with $\mathbb{1}_m$ a length $m$ vector containing ones). We will use $\|\cdot\|$ to denote the operator norm, $\boldsymbol{X}^\dagger$ to denote the pseudo-inverse of $\boldsymbol{X}$, and $\boldsymbol{I}_d$ to denote the $d \times d$ identity matrix.

**Overview.** Although unsupervised alignment is challenging due to the presence of local minima, the imposition of additional structure will help to prune them away. Our key insight is that hierarchical structure decomposes a complicated optimization surface into simpler ones that are less prone to local minima. We formulate a hierarchical Wasserstein approach to align datasets with known (or estimated) clusters $\{\mu_i\}_{i=1}^S, \{\nu_j\}_{j=1}^S$ but whose correspondences are unknown. The task therefore is to jointly learn the alignment $T$ and the cluster-correspondences:

$$\min_{\boldsymbol{P} \in \mathcal{B}_S, T \in \mathcal{T}} \sum_{i=1}^S \sum_{j=1}^S P_{ij} \mathcal{W}_2^2(T(\mu_i), \nu_j), \qquad (2)$$

where the matrix $\boldsymbol{P}$ encodes the strength of correspondences between clusters, with a large $P_{ij}$ value indicating a correspondence between clusters $i, j$, and a small value indicating a lack thereof. We note that $\mathcal{B}_S := \mathcal{U}(S, S)$ is a special type of transport polytope known as the $S$-th Birkhoff polytope. Interestingly, this becomes a nested (or block) OT formulation, where correspondences are resolved at two levels: the outer level resolves cluster-correspondences (via $\boldsymbol{P}$) while the inner level resolves point-wise correspondences between cluster points (via the Wasserstein distance).

**Alignment over the Stiefel manifold.** Assuming clusters lie on subspaces and principal angles between subspaces are "well preserved" across $\boldsymbol{X}$ and $\boldsymbol{Y}$ (we make this precise in Theorem 4.2), an

isometric transformation suffices. Hence, we solve (2) with $\mathcal{T} \leftarrow \mathcal{V}_{D,D}$, the Stiefel manifold which is defined as $\mathcal{V}_{k,d} := \{\boldsymbol{R} \in \mathbb{R}^{k \times d} : \boldsymbol{R}^\top \boldsymbol{R} = \boldsymbol{I}_d\}$. Explicitly, we can re-formulate equation (2) as:

$$\min_{\boldsymbol{P}, \boldsymbol{R}, \{\boldsymbol{Q}_{ij}\}} \sum_{i,j} P_{ij} C_{ij}(\boldsymbol{R}, \boldsymbol{Q}_{ij}) \quad \text{s.t.} \quad \boldsymbol{P} \in \mathcal{B}_S, \quad \boldsymbol{R} \in \mathcal{V}_{D,D}, \quad \boldsymbol{Q}_{ij} \in \mathcal{U}(n_{x,i}, n_{y,j}), \quad (3)$$

$$\text{where} \quad C_{ij}(\boldsymbol{R}, \boldsymbol{Q}_{ij}) := \frac{1}{D} \sum_{k,l} \boldsymbol{Q}_{ij}(k,l) \|\boldsymbol{R}\boldsymbol{X}_i(k) - \boldsymbol{Y}_j(l)\|_2^2 \quad (4)$$

measures pairwise cluster divergences using the squared 2-Wasserstein distance under a Stiefel transformation $\boldsymbol{R}$ acting on the $i^{\text{th}}$ cluster.

Finally, we include entropic regularization over transportation couplings $\boldsymbol{P}$ and all $\boldsymbol{Q}_{ij}$'s to modify the Wasserstein distances to Sinkhorn distances, so as to take advantage of its superior computational and sample complexities. Omitting constraints for brevity, our final problem is given as

$$\min_{\boldsymbol{P}, \boldsymbol{R}, \{\boldsymbol{Q}_{ij}\}} \sum_{i,j} \Big( P_{ij} C_{ij}(\boldsymbol{R}, \boldsymbol{Q}_{ij}) + H_{\gamma_2}(\boldsymbol{Q}_{ij}) \Big) + H_{\gamma_1}(\boldsymbol{P}), \quad (5)$$

where $\gamma_1, \gamma_2 > 0$ are the entropic regularization parameters and the negative entropy function is defined as $H_\gamma(\boldsymbol{P}) := \gamma \sum_{i,j} P_{ij} \log P_{ij}$. Parameters $\gamma_1, \gamma_2$ control the correspondence entropy, therefore (5) approximates (3) when $\gamma_1, \gamma_2 > 0$, but reverts to the original problem (3) as $\gamma_1, \gamma_2 \to 0$.

**Distributed ADMM approach.** Problem (5) is non-convex due to multilinearity in the objective and its Stiefel manifold domain. Although alternating directions method of multipliers (ADMM) is a convergent convex solver framework [45, 46], it is being applied in increasingly many non-convex settings [47]. Since (5) readily admits a splitting structure that separates the individual $C_{ij}$ blocks, we develop a distributed ADMM approach. We proceed to split (5) as follows:

$$\min_{\boldsymbol{P}, \widetilde{\boldsymbol{R}}, \{\boldsymbol{R}_{ij}, \boldsymbol{Q}_{ij}\}} \sum_{i,j} \Big( P_{ij} C_{ij}(\boldsymbol{R}_{ij}, \boldsymbol{Q}_{ij}) + H_{\gamma_2}(\boldsymbol{Q}_{ij}) \Big) + H_{\gamma_1}(\boldsymbol{P}) \quad \text{s.t.} \quad \boldsymbol{R}_{ij} = \widetilde{\boldsymbol{R}}, \;\; \forall i, j,$$

noting that the set constraints are omitted for brevity. The augmented Lagrangian is given by

$$\mathcal{L}_\mu = \sum_{i,j} \Big( P_{ij} C_{ij}(\boldsymbol{R}_{ij}, \boldsymbol{Q}_{ij}) + \langle \frac{\mu}{D} \boldsymbol{\Lambda}_{ij}, \boldsymbol{R}_{ij} - \boldsymbol{R} \rangle + \frac{\mu}{2D} \|\boldsymbol{R}_{ij} - \widetilde{\boldsymbol{R}}\|_F^2 + H_{\gamma_2}(\boldsymbol{Q}_{ij}) \Big) + H_{\gamma_1}(\boldsymbol{P}),$$

where $\mu > 0$ is the ADMM parameter and $\{\Lambda_{ij}\}$ are Lagrange multipliers. Full details of the update steps are included in the Supplementary Material. The algorithm may be summarized in two steps (Alg. 1): (i) a distributed step that asks all cluster pairs to individually find their optimal transformations $\boldsymbol{R}_{ij}$ in parallel, and (ii) a consensus step that aggregates all the locally estimated transformations according to a weighting that is proportional to correspondence strengths $P_{ij}$.

*Parameters.* Entropic parameters $\gamma_1, \gamma_2$ relax the one-to-one cluster correspondence assumption, balancing a trade off between alignment precision (small $\gamma$) and sample complexity (large $\gamma$). Numerically, negative entropy adds strong convexity to the program, reducing sensitivity towards perturbations at the cost of a slower convergence rate. The ADMM parameter $\mu$ controls the 'strength' of the consensus, or from an algorithmic viewpoint, the gradient step size.

*Distributed consensus.* Update steps for $\boldsymbol{Q}_{ij}, \boldsymbol{R}_{ij}, \boldsymbol{L}_{ij}$ can be performed in parallel over all cluster pairs ($S^2$ in total), making it amenable for a distributed implementation. The runtime complexity of this algorithm is presented in the supplementary Materials.

*Robustness against initial conditions.* We intentionally build robustness against initial conditions by ordering updates for $\boldsymbol{R}_{ij}$ and $\boldsymbol{Q}_{ij}$ before $\boldsymbol{P}$ such that when $\mu$ is sufficiently small, the ADMM sequence is influenced more by the data than by initial conditions.

## 4 Theoretical guarantees for cluster-based alignment

While the previous section explains *how* to align clustered datasets, in this section, we aim to answer the question of *when* and *how well* they can be aligned. We provide necessary conditions for *cluster-based alignability* as well as *alignment perturbation bounds* according to equation (3)'s formulation. To simplify our analysis, we make the following assumptions: (i) each of the clusters contain the same number of datapoints $n$, (ii) the ground truth cluster correspondences are $\boldsymbol{P}^\star = \boldsymbol{I}_S / S$ (i.e.,

---

**Algorithm 1** Hierarchical Wasserstein Alignment (HiWA) Algorithm

---

1: **procedure** HIERARCHICALWASSERSTEINALIGNMENT($\gamma_1, \gamma_2, \mu, \{\boldsymbol{X}_i\}_{i=1}^S, \{\boldsymbol{Y}_j\}_{j=1}^S$)
2:     $\boldsymbol{R} \leftarrow$ random $\mathcal{V}_{D,D}$,    $\boldsymbol{P} \leftarrow \mathbb{1}_S \mathbb{1}_S^\top / S^2$,    $\boldsymbol{\Lambda}_{ij} \leftarrow \boldsymbol{0}$,  $\forall i, j$            ▷ Initialization
3:     **while** not converged **do**
4:         **for** all $i, j$ in parallel **do**
5:             $\boldsymbol{Q}_{ij} \leftarrow \mathbb{1}_{n_{x,i}} \mathbb{1}_{n_{y,j}}^\top / n_{x,i} n_{y,j}$
6:             **while** not converged **do**
7:                 $\boldsymbol{R}_{ij} \leftarrow$ STIEFELALIGNMENT($2 P_{ij} \boldsymbol{Y}_j \boldsymbol{Q}_{ij}^\top \boldsymbol{X}_i^\top + \mu(\boldsymbol{R} - \boldsymbol{\Lambda}_{ij})$)
8:                 $\boldsymbol{Q}_{ij} \leftarrow$ SINKHORN($\gamma_2 / P_{ij}, \boldsymbol{C}(k,l) \leftarrow \frac{1}{D} \|\boldsymbol{R}_{ij} \boldsymbol{X}_i(k) - \boldsymbol{Y}_j(l)\|_2^2$)
9:             **end while**
10:         **end for**
11:     $\boldsymbol{P} \leftarrow$ SINKHORN($\gamma_1, \boldsymbol{C}(i,j) \leftarrow C_{ij}(\boldsymbol{R}_{ij}, \boldsymbol{Q}_{ij})$)
12:     $\boldsymbol{R} \leftarrow$ STIEFELALIGNMENT($\sum_{i,j} \boldsymbol{R}_{ij} + \boldsymbol{\Lambda}_{ij}$)
13:     $\boldsymbol{\Lambda}_{ij} \leftarrow \boldsymbol{\Lambda}_{ij} + \boldsymbol{R}_{ij} - \boldsymbol{R}$,    $\forall i, j$
14:     **end while**
15: **end procedure**

---

1: **procedure** SINKHORN($\gamma, \boldsymbol{C} \in \mathbb{R}^{m \times n}$)
2:     $\boldsymbol{K} \leftarrow \exp(-\boldsymbol{C}/\gamma)$,    $\boldsymbol{v} \leftarrow \frac{\mathbb{1}_n}{n}$
3:     **while** not converged **do**
4:         $\boldsymbol{u} \leftarrow \frac{\mathbb{1}_m}{m} \oslash \boldsymbol{K} \boldsymbol{v}$
5:         $\boldsymbol{v} \leftarrow \frac{\mathbb{1}_n}{n} \oslash \boldsymbol{K}^\top \boldsymbol{u}$
6:     **end while**
7:     $\boldsymbol{P} \leftarrow \operatorname{diag}(\boldsymbol{u}) \boldsymbol{K} \operatorname{diag}(\boldsymbol{v})$
8: **end procedure**

1: **procedure** STIEFELALIGNMENT($\boldsymbol{A}$)
2:     $(\boldsymbol{U}, \boldsymbol{\Sigma}, \boldsymbol{V}) \leftarrow \operatorname{SVD}(\boldsymbol{A})$
3:     $\boldsymbol{R} \leftarrow \boldsymbol{U}\boldsymbol{V}^\top$
4: **end procedure**

**Notation:**
$\oslash$: elementwise division
$\exp(\cdot)$: elementwise exponential
$\operatorname{diag}(\cdot)$: diagonal matrix of argument

---

diagonal containing $1/S$). However, this analysis can be extended to the case where the number of points is unequal without loss of generality. Detailed proofs are given in the Supp. Material.

The following result is a criterion that, if met, ensures the existence of a global minimizer of the cluster-correspondence $\boldsymbol{P}^\star$. This criterion requires that matched clusters must be closer in Wasserstein distance than mismatched clusters, according to a threshold determined by Wasserstein's sample complexity (i.e., an asymptotic rate dependent on the clusters' *sample sizes* and *intrinsic dimensions*). Since these sample complexity results are based on the Wasserstein distance, we expect a less stringent criterion when using the Sinkhorn distance in (5) (due to superior sample complexity [37]).

**Theorem 4.1** (Correspondence disambiguity criterion). *Let all clusters be strictly low-rank where the dimension of the $i$-th cluster in the $x$-th dataset is $d_{x,i}$. Let $d_{x,i}, d_{y,j} > 4, \forall i, j \in [\![S]\!]$. Define $\widehat{C}_{ij}^\star :=$ $\min_{\boldsymbol{R} \in \mathcal{V}_{D,D}, \boldsymbol{Q}_{ij} \in \mathcal{B}_n} C_{ij}(\boldsymbol{R}, \boldsymbol{Q}_{ij})$. Problem (3) yields the solution $\boldsymbol{P}^\star = \boldsymbol{I}_S / S$ with probability at least $1 - \delta$ if, $\forall i, j : i \neq j$, the following criterion is satisfied:*

$$\widehat{C}_{ij}^\star + \widehat{C}_{ji}^\star - \widehat{C}_{ii}^\star - \widehat{C}_{jj}^\star > B_{x,i}(\delta) + B_{y,i}(\delta) + B_{x,j}(\delta) + B_{y,j}(\delta)$$

*where $B_{z,k}(\delta) := c_{z,k} n^{-\frac{2}{d_{z,k}}} + \sqrt{\log(1/\delta)/2n}$,    $c_{z,k} = 1458\left(2 + \frac{1}{3^{d_{z,k}/2-2} - 1}\right)$.*

*Proof sketch.* The proof contains two parts. In the first part, we consider perturbation conditions of the cost matrix $\boldsymbol{C}$ in a (non-variational) optimal transport program over the Birkhoff polytope. To be unperturbed from $\boldsymbol{P}^\star = \boldsymbol{I}_S / S$, we require that $C_{ij} + C_{ji} - C_{ii} - C_{jj} > 0, \forall i, j : i \neq j$. In the second part, we extend this condition to the the finite-sample regime by utilizing recently developed concentration bounds [35] for the $p$-Wasserstein distance, which essentially raises the disambiguity lower bound due to finite-sample uncertainty. (Supp. Material, Section 2)    □

Now, even if we know the global correspondence $\boldsymbol{P}^\star$, we still do not have the full picture about the alignment's quality. For example, all matching clusters may have very similar covariances, but principal angles between the clusters are "distorted" across the datasets. Our next theorem gives us an upper bound on the alignment error (for unitary transformations), and makes precise the notion of *global structure distortion*.

**Theorem 4.2** (Cluster-based alignment perturbation bounds). *Consider data matrices $\{\boldsymbol{X}_i, \boldsymbol{Y}_i \in \mathbb{R}^{D \times n}\}_{i=1}^c$ with known point-wise correspondence matrices $\{\boldsymbol{Q}_{ii} \in \mathcal{B}_n\}_{i=1}^c$. Define matrices*

$$\boldsymbol{X} := [\boldsymbol{X}_1 \boldsymbol{Q}_{11}, \boldsymbol{X}_2 \boldsymbol{Q}_{22}, \ldots, \boldsymbol{X}_c \boldsymbol{Q}_{cc}], \qquad \boldsymbol{Y} := [\boldsymbol{Y}_1, \boldsymbol{Y}_2, \ldots, \boldsymbol{Y}_c].$$

*Set $\varepsilon^2 := \left\| \boldsymbol{Y}^\top \boldsymbol{Y} - \boldsymbol{X}^\top \boldsymbol{X} \right\|_F$. If the criterion stated in theorem 4.1 is satisfied, $\boldsymbol{X}$ is full row rank, and $\varepsilon \|\boldsymbol{X}^\dagger\| \le \frac{1}{\sqrt{2}} (\|\boldsymbol{X}\| \, \|\boldsymbol{X}^\dagger\|)^{-1/2}$, then*

$$\min_{\boldsymbol{P} \in \mathcal{B}_c, \boldsymbol{R} \in \mathcal{V}_{D,D}} \sum_{i,j} P_{ij} C_{ij}(\boldsymbol{R}) \quad \le \quad (\|\boldsymbol{X}\| \, \|\boldsymbol{X}^\dagger\| + 2)^2 \|\boldsymbol{X}^\dagger\|^2 \varepsilon^4 + D,$$

*where $D = \sum_{i=1}^c \mathrm{tr}(\boldsymbol{X}_i(\boldsymbol{I}/n - \boldsymbol{Q}_{ii}\boldsymbol{Q}_{ii}^\top)\boldsymbol{X}_i^\top + (1/n - 1)\boldsymbol{Y}_i\boldsymbol{Y}_i^\top)$ is a data-dependent constant.*

*Proof sketch.* We utilize a recent perturbation result on the Procrustes problem (on a Frobenius norm objective) by Arias-Castro et al. [48] and adapt it to our squared 2-Wasserstein objective. (Supp. Material, Section 3) □

Note that $\varepsilon$ plays a major role in the alignment error bound and quantifies the notion of *global structure distortion*, which allows us to understand on how phenomena like covariate shift or misclustering impacts alignment. To shed some light in this regard, we consider a simple analysis on a cluster-pair's error contribution to $\varepsilon$, denoted as $\varepsilon_{ij}$. Consider the decomposition of the $(i,j)$-th block of the Gramians related to clusters $i$ and $j$, where their respective singular value decompositions are $\boldsymbol{X}_i \boldsymbol{Q}_{ii} = \boldsymbol{A}_i \boldsymbol{\Sigma}_{x,i} \boldsymbol{V}^\top$ and $\boldsymbol{Y}_j = \boldsymbol{B}_j \boldsymbol{\Sigma}_{y,j} \boldsymbol{V}^\top$. Defining the *blockwise* error between clusters $i, j$ as

$$\varepsilon_{ij} := \left\| \boldsymbol{Y}_i^\top \boldsymbol{Y}_j - \boldsymbol{Q}_{ij}^\top \boldsymbol{X}_i^\top \boldsymbol{X}_j \boldsymbol{Q}_{jj} \right\|_F = \left\| \boldsymbol{\Sigma}_{y,i} \boldsymbol{B}_i^\top \boldsymbol{B}_j \boldsymbol{\Sigma}_{y,j} - \boldsymbol{\Sigma}_{x,i} \boldsymbol{A}_i^\top \boldsymbol{A}_j \boldsymbol{\Sigma}_{x,j} \right\|_F,$$

two components stand out: (i) *angular shift*, which is characterized by differences in principal angles between $\boldsymbol{B}_i^\top \boldsymbol{B}_j$ and $\boldsymbol{A}_i^\top \boldsymbol{A}_j$, and (ii) *spectral shift*, which is characterized by differences in spectra.

Finally, we show that the subspace configuration of a dataset's clusters can also affect alignment. Pretend for a moment that external alignment information were present to aid in the disambiguation between two clusters. The following lemma tells us when such information is useless (Proof in Supp. Material, Section 4).

**Lemma 4.3** (Uninformative alignment). *Consider clusters $\boldsymbol{X}_i, \boldsymbol{Y}_j \in \mathbb{R}^{D \times n}$ and known point-wise correspondences $\boldsymbol{Q}_{ij} \in \mathcal{U}(n,n)$. Denote the left and right singular vectors of $\boldsymbol{Y}_j \boldsymbol{Q}_{ij}^\top \boldsymbol{X}_i^\top$ associated with the non-zero singular values as $\tilde{\boldsymbol{U}}, \tilde{\boldsymbol{V}} \in \mathbb{R}^{D \times r}$ with $r \le D$. Define the set of orthogonal transformations that are constrained to agree with known angular directions as*

$$\mathcal{T}(\boldsymbol{U}', \boldsymbol{V}') := \{ \boldsymbol{R} \in \mathbb{R}_+^{D \times D} : \boldsymbol{R}^\top \boldsymbol{R} = \boldsymbol{I}, \boldsymbol{R}\boldsymbol{V}' = \boldsymbol{U}' \},$$

*where $\boldsymbol{U}', \boldsymbol{V}' \in \mathcal{V}_{D,r}$ with $r \le D$. Given $\boldsymbol{U}', \boldsymbol{V}' \in \mathbb{R}^{D \times r'}$ with $r' \le D$, we have*

$$\min_{\boldsymbol{R} \in \mathcal{T}(\boldsymbol{U}', \boldsymbol{V}')} C_{ij}(\boldsymbol{R}) \ge \min_{\boldsymbol{R} \in \mathcal{V}_{D,D}} C_{ij}(\boldsymbol{R}), \tag{6}$$

*with equality holding when $\langle \tilde{\boldsymbol{U}}, \boldsymbol{U}' \rangle = \langle \tilde{\boldsymbol{V}}, \boldsymbol{V}' \rangle$.*

Direct consequences of this lemma are the following: When a dataset has equally-spaced subspaces, it has a maximally uninformative geometric configuration since angular information from other clusters (i.e., $\boldsymbol{U}', \boldsymbol{V}'$) can never increase the inter-cluster distance $C_{ij}$ (i.e., equality in (6) always holds); it is hence a worst-case scenario for alignment. This also explains why alignment in very high-dimensional space is harder: All subspaces may be orthogonal to each other, and hence offer no "geometric" advantage.

## 5 Numerical experiments

### 5.1 Synthetic low-rank Gaussian mixture dataset

In this section, we validate our method as well as demonstrate its limiting characteristics under symmetric-subspace and finite-sample regimes. To generate our synthetic data, we repeat the following procedure for each of the $S$ clusters. We first randomly generate Gaussian distribution parameters $\mu_i \in \mathbb{R}^d, \Sigma_i \in \mathbb{R}^d : \Sigma_i \succeq 0$ (positive semi-definite), then randomly sample $n$ data-points from these parameters, and finally project them into a random subspace $\boldsymbol{V}_i \in \mathbb{R}^{D \times d}$ in a $D > d$ dimensional embedding. In these experiments, we assume that the clusters are known, but the

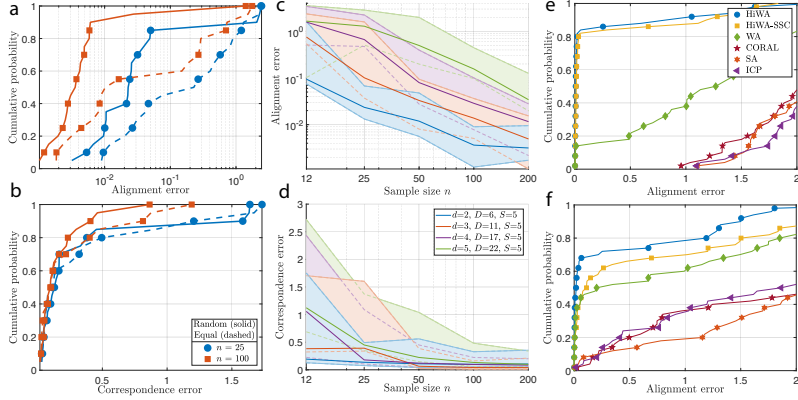

Figure 1: *Synthetic experiments*. HiWA was tested in two subspace configurations (a,b): randomly-spaced (average-case, solid) versus equally-spaced (worst-case, dashed) for $S = 5, d = 2, D = 6, n = \{25, 100\}$, where $S$ is the number of clusters, $d$ the dimension of each cluster, $D$ is the embedding dimension, and $n$ is the sample size. As we expect, performance in terms of the (a) alignment and (b) correspondence error is better in the average (vs. worst) case. In (c,d), we report (c) alignment and (d) correspondence errors as $d$ and $n$ varies, and report the error's $25^{th}/50^{th}/75^{th}$ percentiles. In (e,f), we show ablation results (50 trials, no random restarts permitted) for semi-supervised HiWA (known clusters), *completely* unsupervised HiWA-SSC (unknown clusters), non-structured Wasserstein alignment (WA), subspace alignment methods (SA [29], CORAL [31]), and iterative closest point (ICP) [49] for $n = 50, d = 2$, and (e) $S = 5, D = 6$, and (f) $S = 2, D = 2$.

cluster-correspondence across datasets is unknown. We measure performance with respect to two metrics: (i) *alignment error*, defined as the relative difference between the recovered versus true rotation acting on the data $\|\widehat{\boldsymbol{R}}\boldsymbol{X} - \boldsymbol{R}^{\star}\boldsymbol{X}\|_F^2 / \|\boldsymbol{R}^{\star}\boldsymbol{X}\|_F^2$, and (ii) *correpondence error*, defined as the sum of absolute differences between the recovered and the true correspondences $\sum_{ij} |\widehat{\boldsymbol{P}} - \boldsymbol{P}^{\star}|_{ij}$.

To understand how global geometry impacts alignment, we applied HiWA in two different settings (Figure 1a-b): (i) a *worst-case setting* where subspaces are equally spaced with a subspace similarity of $\|\boldsymbol{V}_i^{\top}\boldsymbol{V}_j\| = 1, \forall i \neq j$, and (ii) the *random setting* where subspaces are randomly selected from the Grassmann manifold. We observe that equally-spaced subspaces have significantly inferior performance when compared to randomly-spaced subspaces, providing some evidence that equally spaced subspaces are indeed the worst-case scenario in alignment, as suggested by Lemma 4.3.

Next, we studied the effect of dimensions $d$ and sample size $n$ on the accuracy of alignment (Figure 1 (c-d)). We tested HiWA across various dataset conditions by varying parameters $d = \{2, 3, 4, 5\}$ and $n = \{12, 25, 50, 100, 200\}$ while approximately maintaining the average subspace correlations (i.e., $\mathbb{E}\|\boldsymbol{V}_i^{\top}\boldsymbol{V}_j\|$) by fixing the cluster size $S = 5$ and tuning $D$ to control the subspace spacing. In both cases, sample complexities are better than the theoretical rate of $O(n^{-1/d})$, which is likely due to the Sinkhorn distance's superior sample complexity. In Figure 1e-f, we conduct an ablation study and evaluate our algorithm against benchmark methods in transfer learning and point set registration in two settings: a simple one in low-$d$ (e) and a harder one in higher-$d$ (f). Specifically, we compare HiWA when clusters are known (but pairwise correspondences are unknown), HiWA with clustering via sparse subspace clustering [12] (HiWA-SSC) to represent *completely* unsupervised alignment, a Wasserstein alignment variant with no cluster-structure (WA) which is akin to OT Procrustes [50, 39, 40, 41], subspace alignment [29], correlation alignment [31], and iterative closest point (ICP) [49]. HiWA exhibits strongest performance, with HiWA-SSC trailing closely behind (since clusters are independently resolved), followed by WA, then other algorithms. Subspace alignment methods have remarkably poor performance in higher dimensions due to their inability to resolve subspace sign ambiguities, while ICP demonstrates its notorious dependence on good initial conditions. These results indicates HiWA's strong robustness against initial conditions and good scaling properties.

### 5.2 Neural population decoding example

Decoding intent (e.g., where you want to move your arm) or evoked responses (e.g., what you are looking at or listening to) directly from neural activity is a widely studied problem in neuroscience, and the first step in the design of a brain machine interface (BMI). A critical challenge with BMIs is that neural decoders need to be recalibrated (or re-trained) due to drift in neural responses or electrophysiology measurements/readouts [51]. A recent method for semi-supervised brain decoding finds a transformation between projected neural responses and movements by solving a KL-divergence

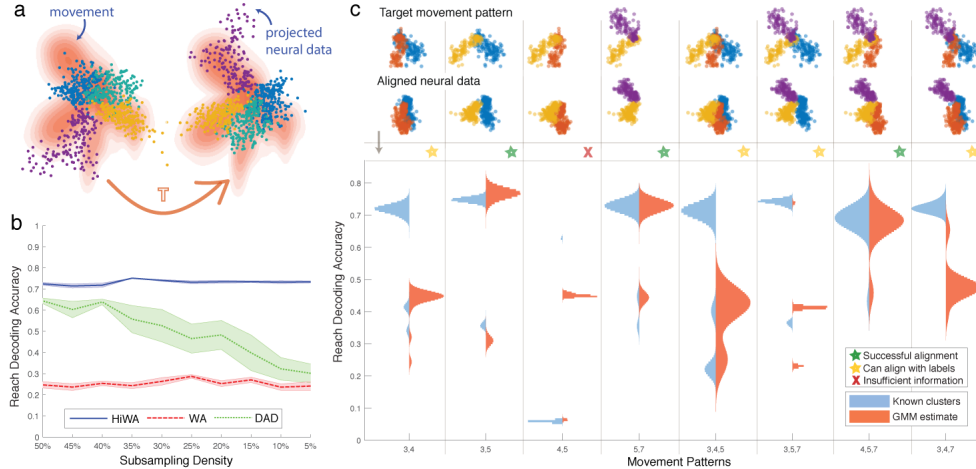

Figure 2: *Results on neural decoding dataset:* How distribution alignment is used to translate neural activity into movement – low-dimensional embeddings of neural data are aligned with target movement patterns (a). In (b), we compare the performance (cluster correspondence) of HiWA, WA, and DAD as the number of points in the source dataset decreases. Next, we compared the performance of HiWA with known and estimated clusters (via GMM). Movement patterns in which cluster separability is high and the geometry is preserved across datasets, can be aligned in both cases (green stars). Patterns where separability is low but geometry is useful can be aligned when the cluster arrangements are known are denoted with yellow stars.

minimization problem [52]. Using this approach, one could build robust decoders that work across days and shifts in neural responses through alignment.

We test the utility of hierarchical alignment for neural decoding on datasets collected from the arm region of primary motor cortex of a non-human primate (NHP) during a center out reaching task [52]. After spike sorting and binning the data, we applied factor analysis to reduce the data dimensionality to 3D (source distribution) and applied HiWA to align the neural data to a 3D movement distribution (target distribution) (Figure 2). We compared its performance to (procrustes) Wasserstein alignment (WA) without hierarchical structure, and a baseline brute force search method called distribution alignment decoding (DAD) [52]. We examined the prediction accuracy of the target reach direction for the motor decoding task (i.e., the cluster classification accuracy).

Next, we examined the impact of the sampling density (Figure 2b) on alignment performance. Our results demonstrate that HiWA continues to produce consistent cluster correspondences (> 70% accuracy), even as the number of samples per cluster drops to 8. In comparison, DAD is competitive at larger sample sizes but its performance rapidly drops off as sampling density decreases because it requires estimating a distribution from samples. WA suffers from the presence of many local minima and fails to find the correct cluster correspondences. Our results suggest that HiWA consistently provides stable solutions, outperforming competitor methods for this application.

Finally, to study the impact of local and global geometry on whether an unlabeled source and target can be aligned, we applied HiWA to permutations of eight subsets of reach directions (movement patterns). When just two reach directions are considered (Figure 2c, Columns 1-4), global geometry becomes useless in determining the correct rotation. In this case, we observe that HiWA is only capable of consistent alignment when cluster asymmetries are sufficiently extreme in both the source and target. When three reach directions are considered (Figure 2c, Columns 5-8), the global geometry can be used, yet there still exist symmetrical cases where recovering the correct rotation is difficult without adequate local asymmetries or some supervised (labeled) data to match clusters. These results suggest that hierarchical structure can be critical in resolving ambiguities in alignment of globally symmetric movement distributions.

## 6 Conclusion

This paper introduces a new method for hierarchical alignment with Wasserstein distances, provided an efficient numerical solution with analytical guarantees. We tested our method and compared its performance against other methods on a synthetic mixture model dataset and on a real neural decoding dataset. Future directions include extensions to non-rigid transformations, and applications to higher dimensional neural datasets that do not rely on external measured behavioral covariates.

**Acknowledgments**

JL was supported by DSO National Laboratories of Singapore, ED and MD were supported by NSF grant IIS-1755871, and CR was supported by NSF grant CCF-1409422 and CAREER award CCF-1350954.

## Footnotes

[3]MATLAB code can be found at https://github.com/siplab-gt/hiwa-matlab. Neural datasets and Python code are provided at http://nerdslab.github.io/neuralign

[4]Dependent on a regularization parameter [37].

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
