[Supplementary Material]

# Hierarchical Optimal Transport for Multimodal Distribution Alignment: Supplementary Material

## 1 ADMM algorithmic details

### 1.1 Derivation of ADMM update steps

ADMM admits the following sequence of updates:

$$(\boldsymbol{R}_{ij}^{(t+1)}, \boldsymbol{Q}_{ij}^{(t+1)}) \leftarrow \underset{\substack{\boldsymbol{R}_{ij} \in S(D,D) \\ \boldsymbol{Q}_{ij} \in \mathcal{U}(n_{x,i}, n_{y_j})}}{\arg\min} \quad P_{ij}^{(t)} C_{ij}(\boldsymbol{R}_{ij}, \boldsymbol{Q}_{ij}) + \frac{\mu}{2D} \|\boldsymbol{R}_{ij} - \boldsymbol{R}^{(t)} + \boldsymbol{\Lambda}_{ij}^{(t)}\|_F^2 + H_{\gamma_2}(\boldsymbol{Q}_{ij}),$$

$$\tag{1}$$

$$\boldsymbol{P}^{(t+1)} \leftarrow \underset{\boldsymbol{P} \in \mathcal{B}_c}{\arg\min} \quad \sum_{i,j} P_{ij} C_{ij}(\boldsymbol{R}_{ij}^{(t+1)}, \boldsymbol{Q}_{ij}^{(t+1)}) + H_{\gamma_1}(\boldsymbol{P}), \tag{2}$$

$$\boldsymbol{R}^{(t+1)} \leftarrow \underset{\boldsymbol{R} \in S(D,D)}{\arg\min} \quad \sum_{i,j} \|\boldsymbol{R}_{ij}^{(t+1)} - \boldsymbol{R} + \boldsymbol{\Lambda}_{ij}^{(t)}\|_F^2, \tag{3}$$

$$\boldsymbol{\Lambda}_{ij}^{(t+1)} \leftarrow \boldsymbol{\Lambda}_{ij}^{(t)} + \boldsymbol{R}_{ij}^{(t+1)} - \boldsymbol{R}. \tag{4}$$

Update (1) involves an alternating minimization over $\boldsymbol{Q}_{ij}$ and $\boldsymbol{R}_{ij}$ whereby the first variable is fixed while the second is minimized, followed by the second fixed and the first minimized, and the procedure is repeated until convergence is achieved. When solving for $\boldsymbol{R}_{ij}$ we have the following Stiefel manifold optimization:

$$\begin{aligned} \boldsymbol{R}_{ij}^{(t+1)} &\leftarrow \underset{\boldsymbol{R}_{ij} \in S(D,D)}{\arg\min} \quad P_{ij}^{(t)} C_{ij}(\boldsymbol{R}_{ij}, \boldsymbol{Q}_{ij}) + \frac{\mu}{2D} \|\boldsymbol{R}_{ij} - \boldsymbol{R}^{(t)} + \boldsymbol{\Lambda}_{ij}^{(t)}\|_F^2 \\ &= \underset{\boldsymbol{R}_{ij} \in S(D,D)}{\arg\max} \quad \operatorname{tr}\left( \left(2 P_{ij}^{(t)} \boldsymbol{Y}_j \boldsymbol{Q}_{ij}^\top \boldsymbol{X}_i^\top + \mu(\boldsymbol{R}^{(t)} - \boldsymbol{\Lambda}_{ij}^{(t)})\right) \boldsymbol{R}_{ij}^\top \right) = \boldsymbol{U}\boldsymbol{V}^\top, \end{aligned}$$

where $2 P_{ij}^{(t)} \boldsymbol{Y}_j \boldsymbol{Q}_{ij}^\top \boldsymbol{X}_i^\top + \mu(\boldsymbol{R}^{(t)} - \boldsymbol{\Lambda}_{ij}^{(t)}) = \boldsymbol{U}\boldsymbol{D}\boldsymbol{V}^\top$ is its SVD. We employ the Sinkhorn algorithm (Algorithm 1 of [1]) to solve for $\boldsymbol{Q}_{ij}$, using an entropic parameter of $\gamma_2/P_{ij}^{(t)}$ and uniform marginals. $\boldsymbol{R}_{ij}^{(t+1)}$ and $\boldsymbol{Q}_{ij}^{(t+1)}$ are retrieved once the alternating minimization converges.

Update (2) also employs the Sinkhorn algorithm over the cost matrix generated by $C_{ij}(\boldsymbol{R}_{ij}^{(t+1)}, \boldsymbol{Q}_{ij}^{(t+1)})$ using variables found in update (1), along with an entropic parameter of $\gamma_1$ and uniform marginals.

Update (3) is a consensus update over a Stiefel manifold optimization:

$$\boldsymbol{R}^{(t+1)} \leftarrow \underset{\boldsymbol{R} \in S(D,D)}{\arg\min} \quad \sum_{i,j} \|\boldsymbol{R}_{ij}^{(t+1)} - \boldsymbol{R} + \boldsymbol{\Lambda}_{ij}^{(t)}\|_F^2 = \boldsymbol{U}\boldsymbol{V}^\top,$$

where $\sum_{i,j} \boldsymbol{R}_{ij}^{(t+1)} + \boldsymbol{\Lambda}_{ij}^{(t)} = \boldsymbol{U}\boldsymbol{D}\boldsymbol{V}^\top$ is its SVD.

## 1.2 Computational complexity of distributed ADMM algorithm

The main computational complexity of the ADMM algorithm comes from line 8, which demands the solving of $S^2$ Sinkhorn problems per ADMM iteration. Fortunately, these Sinkhorn problems (i.e., update steps for $\boldsymbol{Q}_{ij}, \boldsymbol{R}_{ij}, \boldsymbol{L}_{ij}$) may be conducted in parallel, making it amenable for a distributed implementation. When fully parallelized, the algorithm has a per-iteration computational complexity of $O(n_i n_j)$, where $n_i, n_j$ refers to the number of points in the largest clusters of $\boldsymbol{X}, \boldsymbol{Y}$ respectively (compared to vanilla Sinkhorn's $O(n_x n_y)$ complexity where $n_x, n_y$ refers to the *total* number of points in respective datasets, assuming $D \ll \max(n_i, n_j)$). In Figure 1, we record run times of two versions of the same algorithm, with and without parallelism, operating on a similar test dataset, at varying cluster sizes $S$ (50 data points per cluster). 10 random trials were conducted and the means and standard deviations were reported (error lines). As expected, the parallel implementation scales better. These results were computed using an i7 intel processor with 6 cores clocked at 2.6 GHz.

Figure 1: *Run time comparison between a parallelized and a non-parallelized implementation:* We varied the number of clusters – the main bottleneck to run time - to compare the runtimes of the parallelized and non-parallelized implementations.

## 1.3 Stopping criteria.

In lines 3 and 6 of Algorithm 1, possible stopping criteria are (i) $\left\| \boldsymbol{R}^{(t+1)} - \boldsymbol{R}^{(t)} \right\|_F \leq \tau$ where the difference is between the current and previous iteration's transformation and $\tau$ is the tolerance, and (ii) $t \leq T$ where $T$ is the maximum number of iterations.

# 2 Proof of Theorem 4.1

## 2.1 Part 1: Geometric perturbations conditions over the Birkhoff polytope

First, we provide the following lemma that illuminates some basic geometrical insights of the general OT cost matrix, whose coupling is in the Birkhoff polytope $\mathcal{B}_c := \{\boldsymbol{P} \in \mathbb{R}^{c \times c} : \boldsymbol{P} \mathbb{1}_c = \boldsymbol{P}^\top \mathbb{1}_c = \mathbb{1}_c\}$. We define the OT program with respect to a cost matrix $\boldsymbol{C}$ as

$$L(\boldsymbol{C}) := \underset{\boldsymbol{P} \in \mathcal{B}_c}{\arg \min} \ \langle \boldsymbol{P}, \boldsymbol{C} \rangle. \tag{5}$$

The following lemma describes the conditions on $\boldsymbol{C}$ for $\boldsymbol{P}$ to remain unperturbed at $\boldsymbol{P}^\star$.

**Lemma 2.1.** *Define the set of vertices on the $c$-th Birkhoff polytope $\mathcal{B}_c$ that are within a $\delta$-ball from $\boldsymbol{P}^\star$ as*

$$\mathcal{P}_\delta(\boldsymbol{P}^\star) = \{\boldsymbol{P} \in \mathcal{B}_c \setminus \{\boldsymbol{P}^\star\} : \|\boldsymbol{P} - \boldsymbol{P}^\star\|_F \leq \delta\}. \tag{6}$$

*Define the set of matrices that denote directions from $\boldsymbol{P}^\star$ to each neighboring vertex $\boldsymbol{P}_i$ as*

$$\mathcal{V}_\delta(\boldsymbol{P}^\star) = \{\boldsymbol{V} \in \mathbb{R}^{c \times c} : \boldsymbol{V} = \boldsymbol{P} - \boldsymbol{P}^\star, \ \boldsymbol{P} \in \mathcal{P}_\delta(\boldsymbol{P}^\star)\}. \tag{7}$$

*For the linear program's solution $\boldsymbol{P}^\star = L(\boldsymbol{C})$ to remain unchanged, $\forall \boldsymbol{V} \in \mathcal{V}_\delta(\boldsymbol{P}^\star)$ for $\delta = 2$, the cost matrix $\boldsymbol{C}$ should satisfy*

$$\langle \boldsymbol{C}, \boldsymbol{V} \rangle > 0. \tag{8}$$

*Proof.* Birkhoff-von Neuman's theorem [2] states that the optimal transport solution must lie on the convex hull of the $c$-th Birkhoff polytope $\mathcal{B}_c$, and that its vertices are in fact permutation matrices. We therefore say that an LP solution $\boldsymbol{P}^\star$ is a vertex on $\mathcal{B}_c$. The outline of this proof is straightforward: so that $\boldsymbol{P}^\star$ remains unchanged, $\boldsymbol{C}$ should not cause $\boldsymbol{P}^\star$ to move to an adjacent edge of the Birkhoff polytope, nor should it cause it to extend beyond its adjacent edge because then, the neighboring vertex would assume the new solution. For the rest of this proof, we shall let $\boldsymbol{P}^\star = \boldsymbol{I}$ without any loss of generality. We define the set of nearest neighbors to $\boldsymbol{P}^\star$, which are simply permutation matrices that can be described as taking $\boldsymbol{P}^\star$ and exchanging any two columns. For a $\boldsymbol{P}$ with any two columns of $\boldsymbol{P}^\star$ exchanged, notice that the difference matrix $\boldsymbol{V} = \boldsymbol{P} - \boldsymbol{P}^\star$ is a symmetric matrix of mostly zeros except for two off-diagonal $+1$ entries and two diagonal $-1$ entries, hence $\|\boldsymbol{P} - \boldsymbol{P}^\star\|_F = 2$. Formally, we describe the set of nearest neighbors with $\mathcal{P}_\delta(\boldsymbol{P}^\star)$ with $\delta = 2$, defined by (6). Next, we define edges that are adjacent to $\boldsymbol{P}^\star$ using the set $\mathcal{V}_\delta(\boldsymbol{P}^\star)$ also with $\delta = 2$, defined by (7). Note that there are $K := \binom{c}{2} - 1 = \frac{c(c-1)}{2} - 1$ neighboring vertices and adjacent edges.

We will now show how perturbing $\boldsymbol{C}$ in just one direction $\boldsymbol{V}$ changes $\boldsymbol{P}^\star$. First, consider a cost matrix that produce $\boldsymbol{P}^\star$, which is defined as $\boldsymbol{C}^\star := \sum_{\boldsymbol{V}_i \in \mathcal{V}_\delta(\boldsymbol{P}^\star)} \boldsymbol{V}_i$, meaning that it is equi-angle from all $\boldsymbol{V} \in \mathcal{V}_\delta(\boldsymbol{P}^\star)$, for $\delta = 2$. By enumerating over all $\boldsymbol{V}$, we may derive $\boldsymbol{C}^\star = \mathbb{1}\mathbb{1}^\top - c\boldsymbol{I}$, along with the fact that $\frac{\langle \boldsymbol{C}^\star, \boldsymbol{V}\rangle}{\|\boldsymbol{C}^\star\|_F \|\boldsymbol{V}\|_F} = \frac{1}{\sqrt{c-1}}$. Define $\boldsymbol{P}_1 = L(\boldsymbol{C}_1)$ with $\boldsymbol{C}_1$ such that $\langle \boldsymbol{V}_1, \boldsymbol{C}_1 \rangle \leq 0$ and $\frac{\langle \boldsymbol{V}_i, \boldsymbol{C}_1 \rangle}{\|\boldsymbol{V}_i\|_F \|\boldsymbol{C}_1\|_F} = \frac{1}{\sqrt{c-1}}, \forall i \in 2, \dots, K$. Define $\boldsymbol{P}_2 = L(\boldsymbol{C}_2)$ with $\boldsymbol{C}_2$ such that $\frac{\langle \boldsymbol{V}_i, \boldsymbol{C}_2 \rangle}{\|\boldsymbol{V}_i\|_F \|\boldsymbol{C}_2\|_F} = \frac{1}{\sqrt{c-1}}$, $\forall i \in [\![K]\!]$. We make the claim that $\boldsymbol{P}_1 = \boldsymbol{P}_2$ and proceed with a proof by contradiction. As mentioned before, $\forall \boldsymbol{V} \in \mathcal{V}_\delta(\boldsymbol{P}^\star)$ for $\boldsymbol{P}^\star = \boldsymbol{I}$, $\delta = 2$ has exactly four non-zero entries, i.e., $V_{ii} = V_{jj} = -1$ and $V_{ij} = V_{ji} = +1$, where $i,j \in [\![c]\!], i \neq j$. Writing out $\langle \boldsymbol{V}, \boldsymbol{C}\rangle$ explicitly, we have $\langle \boldsymbol{V}, \boldsymbol{C}\rangle = -(C_{ii} + C_{jj}) + (C_{ij} + C_{ji})$. To ensure that $\boldsymbol{C}_1$ and $\boldsymbol{C}_2$ does not interact with other edges of the polytope, we fix $\langle \boldsymbol{V}_i, \boldsymbol{C}\rangle = \langle \boldsymbol{V}_i, \boldsymbol{C}^\star \rangle = c$ for $i = 2, \dots, K$. Since we constructed $\boldsymbol{C}_1$ and $\boldsymbol{C}_2$ to differ only by the condition $\langle \boldsymbol{V}_1, \boldsymbol{C}_1 \rangle \leq 0$ or $\langle \boldsymbol{V}_1, \boldsymbol{C}_2 \rangle > 0$, and any $\boldsymbol{V}_i$ affects only four entries of the cost matrix, we may greatly simplify our analysis of $\boldsymbol{C}$ and $\boldsymbol{P}$ to only these four entries. As such, we extract these four entries of $\boldsymbol{P}$, represent it using $\hat{\boldsymbol{P}} \in \mathcal{B}_2$, and parameterize using $t \in [0,1]$ it as

$$\hat{\boldsymbol{P}}(t) = t\begin{bmatrix} 1 & 0 \\ 0 & 1 \end{bmatrix} + (1-t)\begin{bmatrix} 0 & 1 \\ 1 & 0 \end{bmatrix}. \tag{9}$$

With this parameterized form, we may reexpress the optimization for $L(\boldsymbol{C})$ as

$$\min_{t \in [0,1]} \ t(C_{ii} + C_{jj}) + (1-t)(C_{ij} + C_{ji}) = \min_{t \in [0,1]} \ \Big(\frac{C_{ii} + C_{jj}}{C_{ij} + C_{ji}} - 1\Big)t$$

The above minimization has three cases. If $C_{ii} + C_{jj} = C_{ij} + C_{ji}$ then there exists no unique solution $t$. If $C_{ii} + C_{jj} > C_{ij} + C_{ji}$ then $t = 0$. If $C_{ii} + C_{jj} < C_{ij} + C_{ji}$ then $t = 1$. The first two cases directly corresponds to $\langle \boldsymbol{V}_1, \boldsymbol{C}_1 \rangle \leq 0$, while the third case corresponds to $\langle \boldsymbol{V}_1, \boldsymbol{C}_2 \rangle > 0$. The fact that $t$ is not consistent between all cases demonstrates a contradiction. Moreover, $t = 1$ produces the solution $\hat{\boldsymbol{P}} = \boldsymbol{I}$, and if this holds for all $\boldsymbol{V} \in \mathcal{V}_\delta(\boldsymbol{P}^\star)$, then all off-diagonal entries must be zero and therefore $\boldsymbol{P}^\star = \boldsymbol{I}$ must be the minimizer for $L(\boldsymbol{C})$. $\qquad\square$

A direct consequence of lemma 2.1 is the following.

**Corollary 2.2.** *The solution to the linear program defined by (5) is $\boldsymbol{P}^\star = \boldsymbol{I}$ if the linear cost matrix $\boldsymbol{C}$ satisfies the following property*

$$C_{ij} + C_{ji} - C_{ii} - C_{jj} > 0, \ \ \forall i,j \in [\![c]\!], i \neq j. \tag{10}$$

*Proof.* Analyzing (8), we observe that any $\boldsymbol{V} \in \mathcal{V}_\delta(\boldsymbol{P}^\star)$ for $\delta = 2$ has only four symmetric non-negative entries, which we condense $\boldsymbol{V}$ and $\boldsymbol{C}$ into $\mathbb{R}^{2 \times 2}$ matrices at these four support locations respectively as $\hat{\boldsymbol{V}} = \begin{bmatrix} -1 & +1 \\ +1 & -1 \end{bmatrix}$ and $\hat{\boldsymbol{C}} = \begin{bmatrix} C_{ii} & C_{ji} \\ C_{ij} & C_{jj} \end{bmatrix}$. It thus follows that an explicit computation produces:

$$\langle \boldsymbol{P}, \boldsymbol{C}\rangle \geq 0 \Rightarrow \left\langle \hat{\boldsymbol{P}}, \hat{\boldsymbol{C}} \right\rangle = (C_{ij} + C_{ji}) - (C_{ii} + C_{jj}) \geq 0.$$

Since the set $\mathcal{V}_\delta(\boldsymbol{P}^\star)$ spans all permutations between $i,j \in [\![c]\!], i \neq j$, we conclude with (10). $\quad\square$

In the variational setting, $C_{ij}(\boldsymbol{R}, \boldsymbol{Q}_{ij})$'s are themselves linearly coupled with $\boldsymbol{R}$ and $\boldsymbol{Q}_{ij}$. The following proposition introduces a trivial criterion on $C_{ij}(\boldsymbol{R}, \boldsymbol{Q}_{ij})$'s to guarantee that $\boldsymbol{P}$ remains unperturbed from $\boldsymbol{P}^\star$. For pedagogical reasons, we shall assume that $n \to \infty$ for this proposition but subsequently relax this.

**Proposition 2.3** (Rotationally invariant disambiguity criterion). *Problem* (3) *yields the solution* $\boldsymbol{P}^\star$ *if*, $\forall i, j : i \neq j$, *the following criterion is satisfied:*

$$\min_{\boldsymbol{R}, \boldsymbol{Q}_{ij}} C_{ij}(\boldsymbol{R}, \boldsymbol{Q}_{ij}) + \min_{\boldsymbol{R}, \boldsymbol{Q}_{ji}} C_{ji}(\boldsymbol{R}, \boldsymbol{Q}_{ji}) - \min_{\boldsymbol{R}, \boldsymbol{Q}_{ii}} C_{ii}(\boldsymbol{R}, \boldsymbol{Q}_{ii}) - \min_{\boldsymbol{R}, \boldsymbol{Q}_{jj}} C_{jj}(\boldsymbol{R}, \boldsymbol{Q}_{jj}) > 0. \quad (11)$$

*Proof.* Consider a set of $c$ clusters where clusters $i, j$ satisfy

$$C_{ij}(\boldsymbol{R}'_{ij}, \boldsymbol{Q}'_{ij}) + C_{ji}(\boldsymbol{R}'_{ji}, \boldsymbol{Q}'_{ji}) - C_{ii}(\boldsymbol{R}', \boldsymbol{Q}'_{ii}) - C_{jj}(\boldsymbol{R}', \boldsymbol{Q}'_{jj}) \leq 0, \quad (12)$$

where

$$(\boldsymbol{R}'_{ij}, \boldsymbol{Q}'_{ij}) := \underset{\boldsymbol{R} \in S(d,d), \boldsymbol{Q}_{ij} \in \mathcal{B}_n}{\arg\min} C_{ij}(\boldsymbol{R}),$$

$$(\boldsymbol{R}'_{ji}, \boldsymbol{Q}'_{ji}) := \underset{\boldsymbol{R} \in S(d,d), \boldsymbol{Q}_{ji} \in \mathcal{B}_n}{\arg\min} C_{ji}(\boldsymbol{R}),$$

$$(\boldsymbol{R}', \boldsymbol{Q}'_{ii}, \boldsymbol{Q}'_{jj}) := \underset{\boldsymbol{R} \in S(d,d), \boldsymbol{Q}_{ii}, \boldsymbol{Q}_{jj} \in \mathcal{B}_n}{\arg\min} C_{ii}(\boldsymbol{R}, \boldsymbol{Q}_{ii}) + C_{jj}(\boldsymbol{R}, \boldsymbol{Q}_{jj}).$$

Since the 2-Wasserstein is a valid metric, its distance between any two clusters must satisfy $C_{ij}(\boldsymbol{R}) \geq 0$ with equality holding if and only if the clusters are exactly similar. If $\boldsymbol{P}^\star = \boldsymbol{I}$, similar clusters are denoted with matching indices, and it must follow that $C_{ii}(\boldsymbol{R}') + C_{jj}(\boldsymbol{R}') = 0$. This implies that (12) must be false since $C_{ij}(\boldsymbol{R}'_{ij}), C_{ji}(\boldsymbol{R}'_{ji}) > 0$ for mismatched clusters. Due to this contradiction, the disambiguity criterion (11) must hold for all cluster pairs $i, j : i \neq j$. $\qquad\square$

This proposition provides a disambiguity criterion, requiring that matched clusters (i.e., $C_{ii}, C_{jj}$) should be more similar than mismatched clusters (i.e., $C_{ij}, C_{ji}$) up to some disambiguity *threshold* (in the case of $n \to \infty$, the threshold is 0). To extend this proposition to the finite-sample regime, we require a higher disambiguity-threshold to offset uncertainty due to finite samples.

## 2.2   Part 2: Disambiguity criterion in the finite-sample regime

We utilize a recent $p$-Wasserstein concentration bound by Weed and Bach [3] that describes finite sample behavior on the Wasserstein distance for data embedded in high-dimensional space, but whose clusters are themselves approximately low-dimensional. We will proceed our analysis with the language of probability measures $\mu$ to make our analysis consistent with [3]. We thus define the equivalent measure analogs as follows.

**Definition 2.4.** *Let clusters* $\boldsymbol{X}_i \in \mathbb{R}^{D \times n_{x,i}}$ *and* $\boldsymbol{Y}_j \in \mathbb{R}^{D \times n_{y,j}}$ *be respectively denoted by empirical measures as*

$$\hat{\mu}_{x,i} := \frac{1}{n_{x,i}} \sum_{k=1}^{n_{x,i}} \delta_{\boldsymbol{X}_i(k)}, \qquad \hat{\mu}_{y,i} := \frac{1}{n_{y,j}} \sum_{k=1}^{n_{y,j}} \delta_{\boldsymbol{Y}_j(k)},$$

*where* $\delta_{\boldsymbol{X}_i(k)}$ *refers to a discrete point located at* $\boldsymbol{X}_i(k)$. *At the limit, we denote the measures as*

$$\mu_{x,i} := \lim_{n_{x,i} \to \infty} \frac{1}{n_{x,i}} \sum_{k=1}^{n_{x,i}} \delta_{\boldsymbol{X}_i(k)}, \qquad \mu_{y,j} := \lim_{n_{y,j} \to \infty} \frac{1}{n_{y,j}} \sum_{k=1}^{n_{y,j}} \delta_{\boldsymbol{Y}_j(k)}.$$

**Definition 2.5.** *Denote a linear transformation* $\boldsymbol{R}$ *applied on the measure as*

$$\boldsymbol{R} \circ \hat{\mu}_{x,i} := \frac{1}{n_{x,i}} \sum_{k=1}^{n_{x,i}} \delta_{\boldsymbol{R}\boldsymbol{X}_i(k)}, \qquad \boldsymbol{R} \circ \mu_{x,i} := \lim_{n_{x,i} \to \infty} \frac{1}{n_{x,i}} \sum_{k=1}^{n_{x,i}} \delta_{\boldsymbol{R}\boldsymbol{X}_i(k)}.$$

*The transformed inter-cluster distance between clusters may thus be denoted as*

$$\hat{C}_{ij}(\boldsymbol{R}) := \mathcal{W}_2^2(\boldsymbol{R} \circ \hat{\mu}_{x,i}, \hat{\mu}_{y,i}), \qquad \tilde{C}_{ij}(\boldsymbol{R}) := \mathcal{W}_2^2(\boldsymbol{R} \circ \mu_{x,i}, \mu_{y,i}).$$

Now, we may proceed to state results from [3]. The following result pertains to the sample complexity of measures $\mu$ in $\mathbb{R}^D$ that are supported on an approximately low-dimensional set in $\mathbb{R}^d$, where $d \ll D$. First we require some definitions.

**Definition 2.6.** *Given a set $S \subseteq \mathbb{R}^d$, let $\mathcal{N}_\varepsilon(S)$ denote the $\varepsilon$-covering number of set $S$, which is defined as the minimum number $m$ of closed balls $B_1, \ldots, B_m$ of diameter $\varepsilon$ such that $S \subseteq \bigcup_{1 \leq i \leq m} B_i$.*

**Definition 2.7.** *For any set $S \subseteq \mathbb{R}^d$, the $\varepsilon$-fattening of $S$ is $S_\varepsilon := \{y : D(y, S) \leq \varepsilon\}$.*

**Proposition 2.8** (Weed and Bach [3], Proposition 16)**.** *Let $S$ be a set that satisfies $\mathcal{N}_{\varepsilon'}(S) \leq (3\varepsilon')^{-d}$ for all $\varepsilon' \leq 1/27$ and for some $d > 2p$. Suppose there exists a positive constant $\sigma$ such that $\mu$ satisfies $\mu(S_\varepsilon) \geq 1 - e^{-\varepsilon^2/2\sigma^2}$ for all $\varepsilon > 0$. If $p \log \frac{1}{\sigma} \geq 1/18$, then for all $n \leq (18p\sigma^2 \log \frac{1}{\sigma})^{-d/2}$,*

$$\mathbb{E}[\mathcal{W}_p^p(\mu, \hat{\mu}_n)] \leq cn^{-p/d},$$

*where $c = 27^p (2 + \frac{1}{3^{d/2-p} - 1})$.*

This proposition states that the degree that $\mu$ is concentrated (as parameterized by $\sigma$) around set $S$ (approximately supported in low-dimensions) affects how "long" (in terms of $n$) we can enjoy the fast convergence rate of $n^{-p/d}$. We will leverage this result to obtain the following theorem on cluster correspondence disambiguity with respect to sample complexity.

**Corollary 2.9.** *Let sets $S_{x,i}$ and $S_{y,j}$ satisfy the conditions for $S$ in proposition 2.8 for some $\sigma_{x,i}, \sigma_{y,j} > 0$ and $d_{x,i}, d_{y,j} > 4$. If $\log \frac{1}{\sigma_{x,i}} \geq \frac{1}{36}$ and $\log \frac{1}{\sigma_{y,j}} \geq \frac{1}{36}$, then for all $n_{x,i} \leq (36\sigma_{x,i}^2 \log \frac{1}{\sigma_{x,i}})^{-d_{x,i}/2}$ and $n_{y,j} \leq (36\sigma_{y,j}^2 \log \frac{1}{\sigma_{y,j}})^{-d_{y,j}/2}$,*

$$\mathbb{E}[|\tilde{C}_{ij}(\boldsymbol{R}) - \hat{C}_{ij}(\boldsymbol{R})|] \leq c_{x,i} n_{x,i}^{-2/d_{x,i}} + c_{y,j} n_{y,j}^{-2/d_{y,j}},$$

*where*

$$c_{z,k} = 729(2 + \frac{1}{3^{d_{z,k}/2-2} - 1}).$$

*Proof.* Denote $\mu_i, \mu_j$ as measures and $\hat{\mu}_i, \hat{\mu}_j$ as their empirical estimates. By the triangle inequality,

$$\mathbb{E}[\mathcal{W}_2^2(\hat{\mu}_i, \hat{\mu}_j)] \leq \mathbb{E}[\mathcal{W}_2^2(\hat{\mu}_i, \mu_i) + \mathcal{W}_2^2(\mu_j, \hat{\mu}_j)]$$
$$\leq \mathbb{E}[\mathcal{W}_2^2(\hat{\mu}_i, \mu_i) + \mathcal{W}_2^2(\mu_i, \mu_j) + \mathcal{W}_2^2(\hat{\mu}_j, \mu_j)]$$
$$\Rightarrow \mathbb{E}[|\mathcal{W}_2^2(\mu_i, \mu_j) - \mathcal{W}_2^2(\hat{\mu}_i, \hat{\mu}_j)|] \leq \mathbb{E}[\mathcal{W}_2^2(\hat{\mu}_i, \mu_i)] + \mathbb{E}[\mathcal{W}_2^2(\hat{\mu}_j, \mu_j)]$$
$$\Rightarrow \mathbb{E}[|\tilde{C}_{ij}(\boldsymbol{R}) - \hat{C}_{ij}(\boldsymbol{R})|] \leq c_{x,i} n_{x_i}^{-2/d_{x,i}} + c_{y,j} n_{y,j}^{-2/d_{y,j}},$$

where the last line is a direct application of definition 2.5 and proposition 2.8. $\square$

**Lemma 2.10.** *Let $S_{x,i}, S_{y,i}, S_{x,j}, S_{y,j}$ be sets that satisfy the conditions for $S$ in proposition 2.8 for some $\sigma_{x,i}, \sigma_{y,i}, \sigma_{x,j}, \sigma_{y,j} > 0$ and $d_{x,i}, d_{y,i}, d_{x,j}, d_{y,j} > 4$. If $\log \frac{1}{\sigma_{x,i}}, \log \frac{1}{\sigma_{y,i}}, \log \frac{1}{\sigma_{x,j}}, \log \frac{1}{\sigma_{y,j}} \geq \frac{1}{36}$, then for all $n_{x,i} \leq (36\sigma_{x,i}^2 \log \frac{1}{\sigma_{x,i}})^{-d_{x,i}/2}$, $n_{y,i} \leq (36\sigma_{y,i}^2 \log \frac{1}{\sigma_{y,i}})^{-d_{y,i}/2}$, $n_{x,j} \leq (36\sigma_{x,j}^2 \log \frac{1}{\sigma_{x,j}})^{-d_{x,j}/2}$, and $n_{y,j} \leq (36\sigma_{y,j}^2 \log \frac{1}{\sigma_{y,j}})^{-d_{y,j}/2}$, cluster correspondences in problem (3) may be disambiguated to achieve $\boldsymbol{P}^\star = \boldsymbol{I}$ when the following criterion is fulfilled for all $i, j \in [\![c]\!] : i \neq j$:*

$$\mathbb{E}[\min_{\boldsymbol{R}} \hat{C}_{ij}(\boldsymbol{R}) + \min_{\boldsymbol{R}} \hat{C}_{ji}(\boldsymbol{R}) - \min_{\boldsymbol{R}} \hat{C}_{ii}(\boldsymbol{R}) - \min_{\boldsymbol{R}} \hat{C}_{jj}(\boldsymbol{R})] > 2(B_{x,i} + B_{y,i} + B_{x,j} + B_{y,j}).$$

*where the constants are defined as*

$$B_{z,k} := c_{z,k} n_{z,k}^{-2/d_{z,k}}, \quad c_{z,k} := 729\left(2 + \frac{1}{3^{d_{z,k}/2-2} - 1}\right).$$

*Proof.* This follows a direct application of criterion (11) and corollary 2.9. $\square$

**Proposition 2.11** (Weed and Bach [3], Proposition 20)**.** *For all $n \geq 0$ and $0 \leq p < \infty$,*

$$\mathbb{P}[\mathcal{W}_p^p(\mu, \hat{\mu}_n) \geq \mathbb{E}\mathcal{W}_p^p(\mu, \hat{\mu}_n) + t] \leq \exp(-2nt^2).$$

**Theorem 2.12.** *Define $\hat{C}_{ij}^{\star} := \min_{\boldsymbol{R}\in S(D,D)} \hat{C}_{ij}(\boldsymbol{R})$. If the conditions in Lemma 2.10 are satisfied then problem* (3) *yields the solution $\boldsymbol{P}^{\star} = \boldsymbol{I}$ with probability at least $1 - \delta$ if, $\forall i,j : i \neq j$, the following criterion is satisfied:*

$$\hat{C}_{ij}^{\star} + \hat{C}_{ji}^{\star} - \hat{C}_{ii}^{\star} - \hat{C}_{jj}^{\star} > 2(B_{x,i}(\delta) + B_{y,i}(\delta) + B_{x,j}(\delta) + B_{y,j}(\delta))$$

*where*

$$B_{z,k}(\delta) := c_{z,k} n_{z,k}^{-\frac{2}{d_{z,k}}} + \sqrt{\log(1/\delta)/2n_{z,k}}, \quad c_{z,k} := 729\Big(2 + \frac{1}{3^{d_{z,k}/2-2}-1}\Big), \quad (13)$$

*where $d_{z,k}$ refers to the intrinsic dimension of the $k$-th cluster from the $z$-th dataset.*

*Proof.* For some measure $\mu$ and its empirical finite-sample estimate $\hat{\mu}_n$, proposition 2.11 may be equivalently stated with the choice of $t = \sqrt{\log\frac{1}{\delta}/2n}$ as:

$$|\mathcal{W}_p^p(\mu, \hat{\mu}_n) - \mathbb{E}\mathcal{W}_p^p(\mu, \hat{\mu}_n)| \leq \sqrt{\log(1/\delta)/2n},$$

holds with at least probability $1 - \delta$. Under the conditions stated in proposition (2.8), and combining its result with the above relation, we have

$$\mathcal{W}_p^p(\mu, \hat{\mu}_n) \leq cn^{-p/d} + \sqrt{\log(1/\delta)/2n},$$

where $c = 27^p(2 + \frac{1}{3^{d/2-p}-1})$. Combining this for all terms in the left-hand side of (13) yields the stated result. $\square$

## 2.3 Putting everything together

The final proof of Theorem 4.1 is a simplified version of Theorem 2.12's since we assert a stronger (but cleaner) exact low-rank assumption to streamline communication. When the data is *exactly* supported in low-dimensions (as opposed to approximately), the $\varepsilon$-fattening disappears (i.e., $\varepsilon \to 0$) thus any positive $\sigma < \varepsilon$ will send $n \to \infty$, implying that the rapid convergence in dimensions $d \ll D$ holds for $n \to \infty$. Hence an identical result holds, with the sole condition that $d > 4$. $\square$

## 3 Proof of Theorem 4.2

We apply a very recent perturbation bound for the Procrustes problem developed by Arias-Castro et al. [4] to subsequently state a cluster-based alignment bound. First, we outline the perturbation bound for the classical Procrustes problem below.

**Theorem 3.1** (Procrustes perturbation bounds, Theorem 1 [4])**.** *Consider short matrices $\boldsymbol{X}, \boldsymbol{Y} \in \mathbb{R}^{d\times n}$ with $d < n$ and $\boldsymbol{X}$ having full rank. Set $\varepsilon^2 = \left\|\boldsymbol{Y}^{\top}\boldsymbol{Y} - \boldsymbol{X}^{\top}\boldsymbol{X}\right\|_p$, where $\|\cdot\|_p$ denotes the Schatten p-norm. Denote the singular value decomposition of $\boldsymbol{X} = \boldsymbol{U\Sigma V}^{\top}$, where $\boldsymbol{\Sigma}$ contains diagonal elements $\sigma_1 \geq \sigma_2 \geq \ldots \sigma_d > 0 = \cdots = 0$, and let $\boldsymbol{X}^{\dagger}$ be the pseudo-inverse of $\boldsymbol{X}$, i.e., $\boldsymbol{X}^{\dagger} = \boldsymbol{U\Sigma}^{\dagger}\boldsymbol{V}^{\top}$, where $\boldsymbol{\Sigma}^{\dagger} = \mathrm{diag}(\sigma_1^{-1}, \sigma_2^{-1}, \ldots, \sigma_d^{-1}, 0, \ldots, 0)$. If $\left\|\boldsymbol{X}^{\dagger}\right\|\varepsilon \leq \frac{1}{\sqrt{2}}(\|\boldsymbol{X}\|\left\|\boldsymbol{X}^{\dagger}\right\|)^{-1/2}$ then*

$$\min_{\boldsymbol{R}\in S(d,d)} \|\boldsymbol{RX} - \boldsymbol{Y}\|_p \leq (\|\boldsymbol{X}\|\left\|\boldsymbol{X}^{\dagger}\right\| + 2)\left\|\boldsymbol{X}^{\dagger}\right\|\varepsilon^2.$$

Directly applying Theorem 3.1 using the Schatten 2-norm (i.e., the Frobenius norm) yields

$$\min_{\boldsymbol{R}\in S(d,d)} \|\boldsymbol{RX} - \boldsymbol{Y}\|_F^2 = \min_{\boldsymbol{R}\in S(d,d)} \sum_{i=1}^{c} \mathrm{tr}(\boldsymbol{Q}_{ii}^{\top}\boldsymbol{X}_i^{\top}\boldsymbol{X}_i\boldsymbol{Q}_{ii} + \boldsymbol{Y}_i^{\top}\boldsymbol{Y}_i - 2\boldsymbol{Y}_i\boldsymbol{Q}_{ii}^{\top}\boldsymbol{X}_i^{\top}\boldsymbol{R}^{\top})$$

$$\leq ((\|\boldsymbol{X}\|\left\|\boldsymbol{X}^{\dagger}\right\| + 2)\left\|\boldsymbol{X}^{\dagger}\right\|\varepsilon^2)^2 := B^2,$$

$$\Rightarrow \max_{\boldsymbol{R}\in S(d,d)} \sum_{i=1}^{c} \mathrm{tr}(2\boldsymbol{Y}_i\boldsymbol{Q}_{ii}^{\top}\boldsymbol{X}_i^{\top}\boldsymbol{R}^{\top}) \geq \mathrm{tr}(\boldsymbol{XX}^{\top} + \boldsymbol{YY}^{\top}) - B^2,$$

where $\varepsilon^2 := \left\|\boldsymbol{Y}^{\top}\boldsymbol{Y} - \boldsymbol{X}^{\top}\boldsymbol{X}\right\|_F$ and under the conditions that $\boldsymbol{X}$ is full rank and $\left\|\boldsymbol{X}^{\dagger}\right\|\varepsilon \leq \frac{1}{\sqrt{2}}(\|\boldsymbol{X}\|\|\boldsymbol{X})^{\dagger}\|)^{-1/2}$. When the criterion given by corollary 2.3 is satisfied, we are guaranteed

cluster-correspondences $\boldsymbol{P}^\star = \boldsymbol{I}$. Therefore $\sum_{ij} P_{ij} C_{ij}(\boldsymbol{R}) = \sum_i C_{ii}(\boldsymbol{R})$. We utilize this to lower bound $\sum_i C_{ii}(\boldsymbol{R})$ as follows:

$$
\min_{\boldsymbol{R} \in S(d,d)} \sum_{i=1}^{c} C_{ii}(\boldsymbol{R})
$$

$$
= \min_{\boldsymbol{R} \in S(d,d)} \sum_{i=1}^{c} \operatorname{tr}(\boldsymbol{X}_i \operatorname{diag}(\boldsymbol{Q}_{ii}\mathbb{1})\boldsymbol{X}_i^\top + \boldsymbol{Y}_i \operatorname{diag}(\boldsymbol{Q}_{ii}^\top \mathbb{1})\boldsymbol{Y}_j^\top - 2\boldsymbol{Y}_i \boldsymbol{Q}_{ii}^\top \boldsymbol{X}_i^\top \boldsymbol{R}^\top)
$$

$$
= \sum_{i=1}^{c} \operatorname{tr}(\boldsymbol{X}_i \operatorname{diag}(\boldsymbol{Q}_{ii}\mathbb{1})\boldsymbol{X}_i^\top + \boldsymbol{Y}_i \operatorname{diag}(\boldsymbol{Q}_{ii}^\top \mathbb{1})\boldsymbol{Y}_j^\top) - \max_{\boldsymbol{R} \in S(d,d)} \sum_{i=1}^{c} \operatorname{tr}(2\boldsymbol{Y}_j \boldsymbol{Q}_{ii}^\top \boldsymbol{X}_i^\top \boldsymbol{R}^\top)
$$

$$
\leq B^2 + \sum_{i=1}^{c} \operatorname{tr}(\boldsymbol{X}_i(\operatorname{diag}(\boldsymbol{Q}_{ii}\mathbb{1}) - \boldsymbol{Q}_{ii}\boldsymbol{Q}_{ii}^\top)\boldsymbol{X}_i^\top + \boldsymbol{Y}_j(\operatorname{diag}(\boldsymbol{Q}_{ii}\mathbb{1}) - \boldsymbol{I})\boldsymbol{Y}_j^\top)
$$

$$
= B^2 + \sum_{i=1}^{c} \operatorname{tr}(\boldsymbol{X}_i(1/n - \boldsymbol{Q}_{ii}\boldsymbol{Q}_{ii}^\top)\boldsymbol{X}_i^\top + \boldsymbol{Y}_j(1/n - \boldsymbol{I})\boldsymbol{Y}_j^\top).
$$

□

## 4 Proof of Lemma 4.3

To ease notation, let $\boldsymbol{A} = \boldsymbol{Y}_j \boldsymbol{Q}_{ij}^\top \boldsymbol{X}_i^\top$, and $\boldsymbol{C} = \frac{1}{n}\boldsymbol{X}_i\boldsymbol{X}_i^\top + \frac{1}{n}\boldsymbol{Y}_j\boldsymbol{Y}_j^\top$. Let $\boldsymbol{A}$ be decomposed by the singular-value decomposition as $\boldsymbol{A} = \tilde{\boldsymbol{U}}\tilde{\boldsymbol{\Sigma}}\tilde{\boldsymbol{V}}^\top$. Lastly, let $\boldsymbol{U} = [\boldsymbol{U}', \boldsymbol{U}'']$ and $\boldsymbol{V} = [\boldsymbol{V}', \boldsymbol{V}'']$, where $\langle \boldsymbol{U}'', \boldsymbol{U}' \rangle = \langle \boldsymbol{V}'', \boldsymbol{V}' \rangle = 0$. Then it follows that

$$
\operatorname{tr}(\boldsymbol{U}^\top \tilde{\boldsymbol{U}}\tilde{\boldsymbol{\Sigma}}\tilde{\boldsymbol{V}}^\top \boldsymbol{V}) \leq \operatorname{tr}(\tilde{\boldsymbol{\Sigma}}) = \operatorname{tr}(\tilde{\boldsymbol{U}}^\top \tilde{\boldsymbol{U}}\tilde{\boldsymbol{\Sigma}}\tilde{\boldsymbol{V}}^\top \tilde{\boldsymbol{V}}) \quad (14)
$$

$$
\operatorname{tr}(\boldsymbol{U}^\top \boldsymbol{A}\boldsymbol{V}) \leq \operatorname{tr}(\tilde{\boldsymbol{U}}^\top \boldsymbol{A}\tilde{\boldsymbol{V}})
$$

$$
\operatorname{tr}\left(\boldsymbol{C} - 2\boldsymbol{U}'^\top \boldsymbol{A}\boldsymbol{V}' - 2\boldsymbol{U}''^\top \boldsymbol{A}\boldsymbol{V}''\right) \geq \operatorname{tr}\left(\boldsymbol{C} - 2\tilde{\boldsymbol{U}}^\top \boldsymbol{A}\tilde{\boldsymbol{V}}\right)
$$

$$
\min_{\substack{\boldsymbol{U}'', \boldsymbol{V}'' \in S(d, d-r): \\ \langle \boldsymbol{U}', \boldsymbol{U}'' \rangle = \langle \boldsymbol{V}', \boldsymbol{V}'' \rangle = 0}} \operatorname{tr}\left(\boldsymbol{C} - 2\boldsymbol{A}\boldsymbol{V}'\boldsymbol{U}'^\top - 2\boldsymbol{A}\boldsymbol{V}''\boldsymbol{U}''^\top\right) \geq \min_{\tilde{\boldsymbol{U}}, \tilde{\boldsymbol{V}} \in S(d,d)} \operatorname{tr}\left(\boldsymbol{C} - 2\boldsymbol{A}\tilde{\boldsymbol{V}}\tilde{\boldsymbol{U}}^\top\right)
$$

$$
\min_{\boldsymbol{R} \in \mathcal{T}(\boldsymbol{U}', \boldsymbol{V}')} C_{ij}(\boldsymbol{R}) \geq \min_{\boldsymbol{R} \in S(d,d)} C_{ij}(\boldsymbol{R}).
$$

What remains is for us to show the condition for equality. From (14), we have that

$$
\operatorname{tr}(\boldsymbol{U}^\top \tilde{\boldsymbol{U}}\tilde{\boldsymbol{\Sigma}}\tilde{\boldsymbol{V}}^\top \boldsymbol{V}) = \operatorname{tr}(\tilde{\boldsymbol{V}}^\top \boldsymbol{V}\boldsymbol{U}^\top \tilde{\boldsymbol{U}}\tilde{\boldsymbol{\Sigma}}) \leq \operatorname{tr}(\tilde{\boldsymbol{\Sigma}}),
$$

with equality holding if $\tilde{\boldsymbol{V}}^\top \boldsymbol{V}\boldsymbol{U}^\top \tilde{\boldsymbol{U}} = \boldsymbol{I}$, implying that $\tilde{\boldsymbol{U}}^\top \boldsymbol{U} = \tilde{\boldsymbol{V}}^\top \boldsymbol{V}$, which imply that

$$
\langle \tilde{\boldsymbol{U}}, \boldsymbol{U}' \rangle = \langle \tilde{\boldsymbol{V}}, \boldsymbol{V}' \rangle, \qquad \langle \tilde{\boldsymbol{U}}, \boldsymbol{U}'' \rangle = \langle \tilde{\boldsymbol{V}}, \boldsymbol{V}'' \rangle,
$$

which are obtained via the substitutions $\boldsymbol{U} = [\boldsymbol{U}', \boldsymbol{U}'']$ and $\boldsymbol{V} = [\boldsymbol{V}', \boldsymbol{V}'']$. □

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

# 5 Additional Figures

In this section, we provide an additional figure to compare HiWA with other competitor methods on the brain decoding example.

Figure 2: *Additional comparisons on brain decoding dataset:* Here, we compared the decoding accuracy in terms of: (a) reach direction decoding or the percentage of time points correctly classified in one of four reach directions, and (b) the instantaneous decoding accuracy measured in terms of their $R^2$ values. In both cases, we compare HiWA with WA, DAD, and 6 other methods that are discussed in the main text and studied in synthetic examples.