[Reviews · NeurIPS 2019]

Reviewer 1



Post-rebuttal update: The authors' response is very thorough and clarifies many of my concerns, mostly those due to what it seems was a misunderstanding of what their baselines were (due to inexact/missing explanations). Conditioned on these being clarified, and relevant background work mentioned and contrasted against (as assured by the authors in the rebuttal) I am updating my score upwards. ---------------------- Summary: This paper proposes a two-step hierarchical formulation of optimal transport that leverages cluster information on the source and target domains in order to improve alignment quality. The method essentially treats this problem as two nested transportation problems: one among the samples in pairs of clusters and one across the clusters themselves. In addition, the problem accounts for (and obtains) a global alignment transformation between the spaces. In order to solve the resulting problem, the authors propose an ADMM approach that splits across cluster pairs and can be solved in parallel for these. Then, the paper provides three important theoretical results aimed at understanding the properties and limitations of the proposed approach, particularly with regards to geometric characteristics in the data. Finally, experimental results are presented one toy and one real task consisting of neural population decoding, both of which seem to show the method performs well in practice. Strengths: - Elegant theoretical guarantees for their cluster-based alignment method, which build upon recent sample complexity results to put this hierarchical notion of OT on firm grounding - Interesting real-world application on a neural population decoding task with promising results - Superbly written paper - argumentation is overall very clear and easy to follow Weaknesses: - Moderate novelty. This paper combines various components proposed in previous work (some of it, it seems, unbeknownst to the authors - see Comment 1): hierarchical/structured optimal transport distances, Wasserstein-Procrustes methods, sample complexity results for Wasserstein/Sinkhorn objectives. Thus, I see the contributions of this paper being essentially: putting together these pieces and solving them cleverly via ADMM. - Lacking awareness of related work (see Comment 1) - Missing relevant baselines and runtime experimental results (Comments 2, 3 and 4) Major Comments/Questions: 1 Related Work. My main concern with this paper is its apparent lack of awareness of two very related lines of work. On the one hand, the idea of defining hierarchical OT distances has been explored before in various contexts (e.g., [5], [6] and [7]), and so has leveraging cluster information for structured losses, e.g. [9] and [10] (note that latter of these relies on an ADMM approach too). On the other hand, combining OT with Procrustes alignment has a long history too (e.g, [1]), with recent successful application in high-dimensional problems ([2], [3], [4]). All of these papers solve some version of Eq (4) with orthogonality (or more general constraints), leading to algorithms whose core is identical to Algorithm 1. Given that this paper sits at the intersection of two rich lines of work in the OT literature, I would have expected some effort to contrast their approach, both theoretically and empirically, with all these related methods. 2. Baselines. Related to the point above, any method that does not account for rotations across data domains (e.g., classic Wasserstein distance) is inadequate as a baseline. Comparing to any of the methods [1]-[4] would have been much more informative. In addition, none of the baselines models group structure, which again, would have been easy to remedy by including at least one alternative that does (e.g., [10] or the method of Courty et al, which is cited and mentioned in passing, but not compared against). As for the neuron application, I am not familiar with the DAD method, but the same applies about the lack of comparison to OT-based methods with structure/Procrustes invariance. 3. Conflation of geometric invariance and hierarchical components. Given that this approach combines two independent extensions on the classic OT problem (namely, the hierarchical formulation and the aligment over the stiefel manifold), I would like to understand how important these two are for the applications explored in this work. Yet, no ablation results are provided. A starting point would be to solve the same problem but fixing the transformation T to be the identity, which would provide a lower bound that, when compared against the classic WA, would neatly show the advantage of the hierarchical vs a "flat" classic OT versions of the problem. 4. No runtime results. Since computational efficiency is one of the major contributions touted in the abstract and introduction, I was expecting to see at least empirical and/or a formal convergence/runtime complexity analysis, but neither of these was provided. Since the toy example is relatively small, and no details about the neural population task are provided, the reader is left to wonder about the practical applicability of this framework for real applications. Minor Comments/Typos: - L53. *the* data. - L147. It's not clear to me why (1) is referred to as an update step here. Wrong eqref? - Please provide details (size, dimensionality, interpretation) about the neural population datasets, at least on the supplement. Many readers will not be familiar with it. References: * OT-based methods to align in the presence of unitary transformations: [1] Rangarajan et al, "The Softassign Procrustes Matching Algorithm", 1997. [2] Zhang et al, "Earth Mover’s Distance Minimization for Unsupervised Bilingual Lexicon Induction", 2017. [3] Alvarez-Melis et al, "Towards Optimal Transport with Global Invariances", 2019. [4] Grave et al, "Unsupervised Alignment of Embeddings with Wasserstein Procrustes", 2019. *Hierarchical OT methods: [5] Yuorochkin et al, "Hierarhical Optimal Transport for Document Representation". [6] Shmitzer and Schnorr, "A Hierarchical Approach to Optimal Transport", 2013 [7] Dukler et al, "Wasserstein of Wasserstein Loss for Learning Generative Models", 2019 [9] Alvarez-Melis et al, "Structured Optimal Transport", 2018 [10] Das and Lee, "Unsupervised Domain Adaptation Using Regularized Hyper-Graph Matching", 2018

Reviewer 2



------------------ Update after reading Author Feedback ----------------- My score remains unchanged. The authors have promised to clarify portions of the text that would more clearly distinguish between their paper and similar works. I believe the paper elegantly merges two interesting ideas in the optimal transport literature, and provides an efficient algorithm for computing a solution to their new problem. ------------------------------------------------------------------------------------- Summary: The authors propose a method for computing Wasserstein distances between distributions with clustered structure. Their hierarchical approach simultaneously solves a cluster alignment problem and a global transport problem. This approach has better robustness properties than typical transport. Comments: I very much liked the paper, and was impressed by the clarity of exposition, the simplicity of the algorithm, and the theoretical details. I have a few comments: On line 133 you mention that ADMM is convergent even in non-convex settings. Reference [38] does show convergence for a few families of problems with non-convex objectives and constraints, but I am unsure whether they apply to your case. I don't fully understand why your augmented Lagrangian is as it's given after line 136. Wouldn't it be \sum_{i, j} (P_{ij}C_{ij}(R_{ij}, Q_{ij} + H_{\gamma_2}(Q_{ij}) + <\Lambda_{ij}, R-R_{ij}> + \frac{\mu}{2D}\|R - R_{ij}\|_F^2) + H_{\gamma_1}(P) Perhaps I am missing a step or a simplification. In lines 142--146 you discuss the effect of the entropy parameter on the algorithm. Can you comment on whether it affects convergence of the StiefelAlignment step? As the regularization parameter increases do you tend to see more oscillation in the transformations?

Reviewer 3



---------------------------- Post rebuttal update ------------------------------- Authors clarified in the rebuttal that they have already been using a variation of a baseline and simulated experimental setup that I was asking for in the review and that misunderstanding arose due to insufficient clarity in some parts of the paper. I believe this paper has some interesting ideas and novel results that are of interest to the community. I am increasing my score, provided authors will include necessary clarifications and discuss missing related works. I’d also appreciate if the code is publicly released. ------------------------------------------------------------------------------------ I have summarized the positive aspects of this paper in the "contributions" section. Here I will focus on the questions, concerns and suggestions for improvement. My key concern is the relation to the earlier work [1] not mentioned in this paper. [1] studies the problem of OT under invariances such as rotation and translation and demonstrate applications on unsupervised word translation. Their formulation is essentially the eq. (4) of this paper. This paper differs from [1] as it allows to account for clustering structure via the hierarchical OT formulation. I think [1] should be considered as one of the key baselines to justify the approach presented in this paper. Application to word translation could serve as a an interesting higher dimensional example increasing significance of this paper. Word embeddings are suitable for clustering (empirically) and there is an optimism that HiWA may work well there. [1] Alvarez-Melis, D., Jegelka, S., & Jaakkola, T. S. (2018). Towards optimal transport with global invariances. arXiv preprint arXiv:1806.09277. Question regarding synthetic experiment: how is the second dataset generated and how is the true R computed? What is the reason to use a different random subspace per cluster (besides verifying Lemma 4.3). Equations (3) and (4) suggest that there is a single unknown orthogonal matrix miss-aligning two datasets. Why not simply generate data from a mixture of Gaussians, then apply a random rotation to parameters of the mixture and generate second dataset - this seems to better align with the problem formulation. Minor comments and typos: 132: augmented -> alternating 156-157: links are not working; what is update (1) and update (2)? 177: program -> problem 179: extra "the"

[Author Response · NeurIPS 2019]

We thank the reviewers for their careful reading and thoughtful feedback regarding our manuscript. Many of these comments will be incorporated in our latest revision. We appreciate the opportunity to clarify some points and address some of the reviewers' concerns, which substantially consist of requests for more extensive *background comparisons* (in text) and *baseline comparisons* (in tests). We will make several changes on revision to address these comments, including adding discussion of more relevant citations and clarifying that we are already performing the appropriate baseline comparisons asked for by the reviewers (which we see was unclear in the writing of our first draft). Most importantly, we will clarify the specific problem space we are targeting as being both 1) completely unsupervised, and 2) requiring non-identity transformations, as is assumed by many other papers in the literature. This clarification will make apparent that we are comparing against all appropriate baselines. We will summarize our contributions as follows and planned revisions to address reviewer comments in the context of these contributions. Please note that we draw our references from R1's citations [1-10] below.

**Contribution 1: our paper is the first to combine OT Procrustes [1-4] with hierarchical OT [5-10]**. Specifically, we perform alignment in an **unsupervised setting where cluster pairing/ordering is not required**, which is a substantially more challenging problem than that addressed by group-based methods such as Courty et al. and [10] (where group labels *are* required). Since our proposed algorithm (e.g., HiWA-SSC as described in Fig. 1e-f) is a *completely* unsupervised method, using R1's suggested group-based *semi-supervised* methods as a baseline comparison is inappropriate. We see how this confusion has arisen from our manuscript and our revision will include a more detailed discussion of these comparisons to substantially clarify the challenging problem space that we uniquely address.

**Contribution 2: novel distributed ADMM numerical algorithm for solving the OT Procrustes and hierarchical OT problems jointly**. Although Algorithm 1 has certain elements of [1-4] (as correctly noted by R1), we make a substantial advance over [1-4] because our work also jointly solves the hierarchical OT problem [5-10] in Eq. (5) (which easily converges to a local minima with naïve approaches). Our proposal of using distributed ADMM to solve the joint problem both effectively finds solutions and is computationally efficient (discussed below). Indeed, [10] also uses ADMM, but in an entirely different way: within each conditional gradients iteration, ADMM is employed to find the correspondence matrix. Unfortunately, the formulation in [10] does not admit a distributed approach. In contrast, our primary approach is ADMM, where splitting (distribution) occurs across all cluster pairs: within each ADMM iteration, we use alternating minimization to find the correspondences (letting us exploit Sinkhorn, which is efficient and fast). It is important to point out that our distributed optimization approach represents a *novel* way of numerically tackling problems in hierarchical OT settings. We will clarify this distinction in the revised document.

**Contribution 3: a novel analysis framework.** We provide a first analysis (specific to our formulation) of the dataset conditions required to solve cluster-based alignment, in addition to providing perturbation and failure mode analyses.

**Ablation studies**. R1 and R4 point to our lack of a baseline comparison against OT Procrustes [1-4] types of methods and recommend an ablation study to test the utility of the hierarchical component of our algorithm. In fact, we **are performing exactly this comparison**, but we see how our imprecise description of the "Wasserstein Alignment (WA)" method (in Fig. 1e-f) has led to this misunderstanding. In revision, we will clarify that WA indeed solves Eq. (4) (similar to methods proposed in [1-4]), serving as an ablation study that jointy finds transformation and correspondences without any cluster structure. We appreciate R1's suggestion of an ablation using just the hierarchical component (with an identity transformation), but we believe this has arisen from a lack of clarity our description of the problem space we target. While identity transformations have been used in cases where target and source domains are already similar (e.g., USPS and MNIST digits of [9, §5.2]), the literature has clearly identified that it is generally required to find non-identity transformations in many cases of interest (e.g., [1-4], especially the discussion in [3]). In the revised manuscript, we will substantially clarify our focus on the setting where invariant transformations are necessary, therefore making a comparison with hierarchical OT (with identity transformations) vacuous.

**Speed/complexity results**. Although we state the runtime (per-iteration) complexity at the end of section 3, no formal derivation was given due to space limitations. Following the suggestion of R1, we will, in the supplementary material of the revision, (i) compare the runtime of our algorithm with and without parallelism, and (ii) give our derivation of the runtime complexity.

**Data generation**. A misunderstanding in the data generation procedure has arisen due to our unclear explanation. We will clarify that we are using exactly the data generation procedure described by R4.

**Error in equation**. We are grateful to R3 for pointing out a typo in our augmented Lagrangian. We were indeed missing an additional $-\frac{\mu}{2D}\|\Lambda_{ij}\|_F^2$ term – expanding our augmented Lagrangian with this additional term would result in a similar form as the one suggested by R3, with only a scaling difference. In our revision, we will express the augmented Lagrangian in the classical form as suggested by R3, with the presence of a $\frac{\mu}{D}$ scaling on the Lagrange multiplier so that Algorithm 1 can remain unchanged.

**ADMM convergence**. We will revise to reflect R3's note that [38] (*our* citation) may not be immediately applicable.

[Meta-Review · NeurIPS 2019]

The paper was originally scored well but with some concerns by the reviewers. After reading the feedback and discussion the reviewers with the lowest score found the feedback very good and incremented their score leading to a clear accept. Please incorporate the very good points in the feedback (more details about what the baselines and discussion with state of the art) in the final paper.